# GENERALIZATION ERROR BOUND VIA EMBEDDING DIMENSION AND NETWORK LIPSCHITZ CONSTANT

## ABSTRACT

Modern deep networks generalize well even in heavily over-parameterized regimes, where traditional parameter-based bounds become vacuous. We propose a representation-centric view of generalization, showing that the generalization error is controlled jointly by: (i) the intrinsic dimension of learned embeddings, which reflects how much the data distribution is compressed and determines how quickly the empirical distribution of embeddings converges to the population distribution in Wasserstein distance, and (ii) the sensitivity of the downstream mapping from embeddings to predictions, quantified by Lipschitz constants. Together these factors yield a new generalization error bound that explicitly links embedding dimension with network architecture. At the final embedding layer, architectural sensitivity vanishes, and the bound is driven more strongly by embedding dimension, explaining why final-layer dimensionality is often a strong empirical predictor of generalization. Experiments across datasets, architectures and controlled interventions validate the theoretical predictions and demonstrate the practical value of embedding-based diagnostics. Overall, this work shifts the focus of generalization analysis from parameter to representation geometry, offering both theoretical insight and actionable tools for deep learning practice.

## 1 INTRODUCTION

Deep networks can generalize effectively even in strongly overparameterized regimes, a phenomenon that remains difficult to explain using classical capacity measures. Traditional bounds based on VC-dimension (Vapnik et al., 1994; Sontag et al., 1998), Rademacher complexity (Truong, 2022), PAC-Bayes theory (Hellström et al., 2025; Lotfi et al., 2022), and algorithmic stability (Feldman & Vondrak, 2018; Alabdulmohsin, 2015) provide valuable insights but often become vacuous at modern scales, as they focus primarily on parameter counts or optimization dynamics rather than the structure of the learned representations.

These limitations have motivated a shift toward studying the *geometry of hidden embeddings*, which reflects the combined influence of data, architecture, and training. Geometric properties such as consistency or separability of representation have been shown to correlate with generalization (Davies & Bouldin, 2009; Dyballa et al., 2024; Belcher et al., 2020), yet many existing metrics rely on labels or capture only local structure, restricting their applicability in settings like self-supervised learning.

A particularly promising direction is the study of *intrinsic dimension*, a label-free measure of the effective degrees of freedom of embeddings. Empirical evidence suggests that lower-dimensional representations generalize better across architectures and training paradigms (Ansuini et al., 2019; Pope et al., 2021), highlighting the need for a theoretical framework that explains this relationship. This motivates our work, which develops a dimension-dependent approach to characterizing representation geometry and its role in generalization.

We address this question by deriving a generalization error bound that makes the role of embedding dimension explicit (Figure 1A). Building on sharp Wasserstein convergence results (Weed & Bach, 2019), we show that for each layer $k$ with intrinsic dimension $d_k$, constants $C_k, D_k$, sensitivity $\bar{L}_k$, population risk $R(F)$ and empirical risk $\hat{R}_n(F)$, the generalization error satisfies

$$R(F) \lesssim \hat{R}_n(F) + \bar{L}_k\left(C_k\, n^{-1/(d_k+\epsilon)} + D_k\sqrt{\tfrac{1}{2n}\log\tfrac{2(L+1)}{\delta}}\right) + \text{(irreducible noise)}.$$

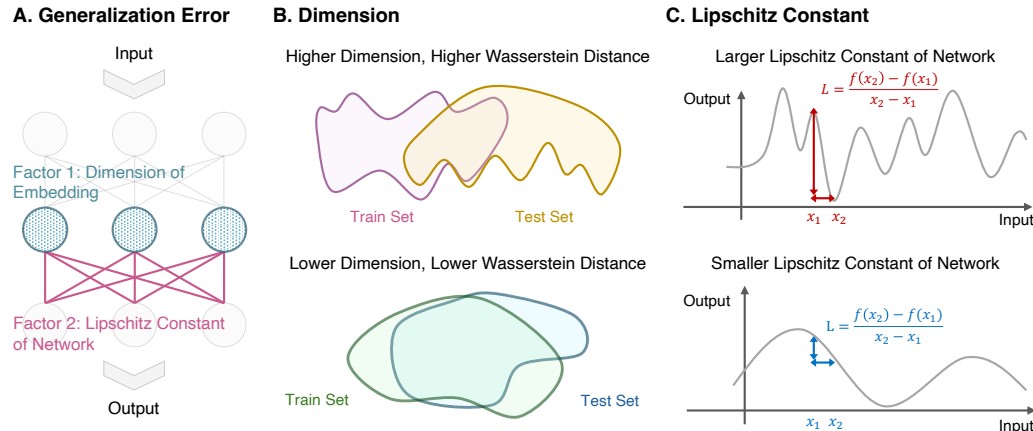

Figure 1: **Embedding Dimension and Lipschitz Constant of Network Jointly Influence Generalization Error.** **(A)** Generalization error depends jointly on embedding dimension and network's Lipschitz constant. **(B)** Lower intrinsic dimension accelerates convergence of empirical to population distribution. **(C)** Smaller Lipschitz constants reduce output sensitivity to perturbations.

Here the $n^{-1/(d_k+\epsilon)}$ term quantifies how quickly the empirical embedding distribution converges to the true population distribution. In other words, with a fixed number of samples, embeddings of lower intrinsic dimension provide a more faithful approximation of their underlying distribution, leading to smaller generalization error (Figure 1B). The factor $\bar{L}_k$ quantifies how perturbations in embeddings propagate through the downstream mappings and ultimately affect the loss (Figure 1C). "Irreducible noise" refers to errors due to label noise or Bayes risk that cannot be reduced by learning.

At the final layer, i.e., the model output, the downstream mapping reduces to the identity, so there is no additional architectural Lipschitz amplification. The resulting bound therefore depends only on the embedding dimension together with data-related constants, such as the loss sensitivity and the Bayes smoothness $L_L(F^*)$:

$$ R(F) \; \lesssim \; \hat{R}_n(F) + (M_F + L_L(F^*)M_{F^*})\Big(C_L\, n^{-1/(d_L+\epsilon)} + D_L\sqrt{\tfrac{1}{2n}\log\tfrac{2(L+1)}{\delta}}\Big) $$
$$ + \text{(irreducible noise)}. $$

where $(M_F + L_L(F^*)M_{F^*})$ is a constant determined by the form of the loss function and bayes predictor. This shows that in final layer, generalization error depends primarily on intrinsic dimension and data-related factors, providing a theoretical explanation for why final-layer dimension is often a strong empirical predictor of generalization.

Our contributions are threefold:

1. We provide a high-probability generalization bound that makes the $n^{-1/(d+\epsilon)}$ dependence on embedding dimension $d$ explicit.
2. We demonstrate that at the final layer the bound simplifies, explaining why final-layer dimension is a strong empirical predictor of generalization.
3. Controlled experiments confirm the predicted scaling and highlight the joint role of dimension and Lipschitz constant of network at intermediate layers.

## 2 RELATED WORKS

**Classical Generalization Bounds.** Theoretical analyses of generalization in deep learning have traditionally centered on parameter-space complexity, including VC-dimension and Rademacher complexity bounds (Sain, 1996; Bartlett & Mendelson, 2002), which provide worst-case guarantees based on the number of parameters. While refinements such as margin-based and norm-based

bounds (Bartlett et al., 2017; Neyshabur et al., 2015; 2017) yield tighter estimates by incorporating weight norms or spectral properties, they become vacuous in the context of modern overparameterized networks. PAC-Bayesian approaches (Arora et al., 2018; Hellström et al., 2025; Lotfi et al., 2022) provide some of the strongest non-vacuous estimates by controlling the complexity of posterior distributions over parameters, often informed by stochastic training dynamics. Stability-based bounds, particularly those grounded in algorithmic stability (Hardt et al., 2016; Feldman & Vondrak, 2018; Alabdulmohsin, 2015), characterize generalization through the sensitivity of a learning algorithm to perturbations in the training data. However, most existing theories are parameter- or algorithm-centric, leaving the role of representation structure in generalization largely unexplored.

**Representation-Based Approaches.** Recent research has increasingly focused on the impact of embedding geometry on generalization. Key approaches analyze properties such as Consistency and Separability of Representations (Davies & Bouldin, 2009; Dyballa et al., 2024; Belcher et al., 2020). These geometric metrics offer improved interpretability and are less reliant on model scale. However, they require labeled data, limiting their applicability in scenarios like pretraining or self-supervised learning, where label information is unavailable.

**Intrinsic Dimension of Representations.** A recent and promising direction in representation-based analysis is the study of *intrinsic dimension*, which quantifies the complexity of embeddings. This approach aligns with the growing understanding that models implicitly compress input data during learning: a lower intrinsic dimension signifies stronger compression and has been empirically linked to improved generalization (Ansuini et al., 2019; Pope et al., 2021). While this offers a quantitative measure of how representations condense information, the theoretical mechanisms connecting intrinsic dimension to generalization remain largely unexplored.

## 3 PRELIMINARIES

We introduce the key concepts and assumptions that connect representation geometry to generalization. Technical variants and detailed proofs are deferred to the appendix.

### 3.1 MEASURES AND WASSERSTEIN DISTANCE

**Definition 1** (Empirical measure). *Let $\mu$ be a probability distribution on a metric space $(X, d)$. Given $n$ i.i.d. samples $\{x_i\}_{i=1}^{n} \sim \mu$, the empirical distribution is*

$$\hat{\mu}_n = \frac{1}{n} \sum_{i=1}^{n} \delta_{x_i}.$$

**Definition 2** (Wasserstein distance). *For two probability measures $\alpha, \beta$ on $(X, d)$, the 1-Wasserstein distance is*

$$\mathcal{W}_1(\alpha, \beta) = \inf_{\gamma \in \Gamma(\alpha, \beta)} \int d(x, y) \, d\gamma(x, y),$$

*where $\Gamma(\alpha, \beta)$ is the set of couplings of $\alpha$ and $\beta$. It can be interpreted as the minimal transport cost between $\alpha$ and $\beta$.*

In our setting, $\mathcal{W}_1$ quantifies how well the empirical embedding distribution $\hat{P}_{k,n}^Z$ approximates its population counterpart $P_k^Z$.

### 3.2 NETWORK DECOMPOSITION AND EMBEDDINGS

**Definition 3** (Network decomposition). *At an intermediate layer $k$, we decompose the network into an encoder $F_{\leq k} : \mathcal{X} \to \mathcal{Z}_k$ mapping the input $x \in \mathcal{X}$ to an embedding $z \in \mathcal{Z}_k$, and a tail map $F_k : \mathcal{Z}_k \to \mathbb{R}^{\overline{C}}$ producing the final prediction. Thus the overall predictor is*

$$F(x) = F_k\big(F_{\leq k}(x)\big).$$

**Definition 4** (Empirical and Population Embedding Distributions). *Given $n$ i.i.d. samples $\{x_i\}_{i=1}^n \sim P_X$, the empirical embedding distribution at layer $k$ is defined as*

$$\hat{\tilde{P}}_{k,n}^Z = \frac{1}{n} \sum_{i=1}^n \delta_{F_{\leq k}(x_i)},$$

*where $F_{\leq k}(x_i)$ represents the embedding of sample $x_i$ at layer $k$.*

*The population embedding distribution $\tilde{P}_k^Z$ is the true distribution of embeddings over the entire data distribution $P_X$, i.e.,*

$$\tilde{P}_k^Z = \mathbb{E}_{x \sim P_X}[\delta_{F_{\leq k}(x)}].$$

**Remarks.** In this work, we use the embeddings of the validation set as a proxy for the empirical embedding distribution and the embeddings of the test set as a proxy for the population embedding distribution. Since the model has not seen the validation or test set during training, this ensures that the validation and test samples remain i.i.d. in the embedding space.

### 3.3 LIPSCHITZ CONTINUITY

**Definition 5** (Lipschitz map). *A function $f : (X, d_X) \to (Y, d_Y)$ is L-Lipschitz if*

$$d_Y(f(x), f(x')) \leq L\, d_X(x, x') \quad \text{for all } x, x' \in X.$$

The Lipschitz constant $L$ measures how perturbations in the input $x$ are amplified in the output $f(x)$.

### 3.4 GEOMETRIC COMPLEXITY AND WASSERSTEIN CONVERGENCE

To state the convergence guarantees of empirical measures in Wasserstein distance, we follow the geometric framework of Weed & Bach (2019). The key idea is that Wasserstein convergence is governed not by the ambient dimension of the space, but by the *intrinsic geometric dimension* of the underlying distribution. We summarize the necessary definitions and present the sharp asymptotic bound.

**Definition 6** (Covering numbers and measure covering dimension). *Let $(X, d)$ be a metric space and $S \subseteq X$. The $\varepsilon$-covering number of $S$ is*

$$\mathcal{N}_\varepsilon(S) := \min \left\{ N : S \subseteq \bigcup_{i=1}^N B_i, \ \text{diam}(B_i) \leq \varepsilon \right\}.$$

*For a probability measure $\mu$ on $X$, the $(\varepsilon, \tau)$-covering number is*

$$\mathcal{N}_\varepsilon(\mu, \tau) := \inf\{\mathcal{N}_\varepsilon(S) : \mu(S) \geq 1 - \tau\},$$

*and the associated $(\varepsilon, \tau)$-dimension is*

$$d_\varepsilon(\mu, \tau) := \frac{\log \mathcal{N}_\varepsilon(\mu, \tau)}{-\log \varepsilon}.$$

**Remarks.** The quantity $d_\varepsilon(\mu, \tau)$ measures the effective geometric complexity of the *bulk* of the measure $\mu$ at scale $\varepsilon$, while allowing a $\tau$-fraction of mass to be ignored.

**Definition 7** (Upper Wasserstein dimension). *For a probability measure $\mu$ on $(X, d)$ and $p \geq 1$, the upper Wasserstein dimension is*

$$d_p^*(\mu) := \inf \left\{ s > 2p \ : \ \limsup_{\varepsilon \to 0} d_\varepsilon\left(\mu,\ \varepsilon^{sp/(s-2p)}\right) \leq s \right\}.$$

**Remarks.** The tolerance parameter $\tau = \varepsilon^{sp/(s-2p)}$ controls the amount of mass that may be discarded at resolution $\varepsilon$, preventing negligible high-complexity regions from dominating the dimension estimate. The value $d_p^*(\mu)$ identifies the smallest exponent $s$ for which the majority of $\mu$ behaves like an $s$-dimensional set at sufficiently fine scales. As shown in Weed & Bach (2019), $d_p^*(\mu)$ is the critical dimension governing the minimax convergence rates for Wasserstein estimation.

**Wasserstein convergence rates.**  The role of $d_p^*(\mu)$ becomes explicit in the convergence behavior of empirical measures. Let $\hat{\mu}_n$ be the empirical distribution of $n$ i.i.d. samples from $\mu$. The following result restates the upper bound of Weed & Bach (2019) in a form that highlights how $d_p^*(\mu)$ controls the rate.

**Theorem 3.1** (Wasserstein convergence governed by the upper Wasserstein dimension). *For any $p \in [1, \infty)$ and any $\varepsilon > 0$, setting $s = d_p^*(\mu) + \varepsilon$ yields the upper bound*

$$\mathbb{E}[W_p(\mu, \hat{\mu}_n)] \leq C_{\varepsilon, p} \, n^{-1/s}.$$

*Since $\varepsilon$ may be chosen arbitrarily small, the empirical Wasserstein convergence rate can be made arbitrarily close to $n^{-1/d_p^*(\mu)}$. Thus $d_p^*(\mu)$ fully determines the asymptotic speed at which empirical measures converge to $\mu$ in Wasserstein distance.*

**Remark 1.** *By taking $s = d_p^*(\mu) + \varepsilon$ in Theorem 3.1, we make the dependence of the convergence rate on the measure's intrinsic dimension explicit: a larger $d_p^*(\mu)$ directly yields a slower rate $n^{-1/s}$. Thus lower-dimensional distributions enjoy faster Wasserstein convergence, while higher-dimensional ones converge more slowly. In representation learning, estimating an intrinsic-dimension proxy from embeddings provides an empirical estimate of $d_p^*(\mu)$ and therefore predicts how efficiently finite samples recover the population representation geometry.*

**Definition 8** (Population Risk and Empirical Risk). *For a predictor $F : \mathcal{X} \to \mathbb{R}^C$ and loss function $\ell$, the* population risk *is*

$$R(F) := \mathbb{E}_{(x,y) \sim P_{X,Y}}[\ell(F(x), y)],$$

*and the* empirical risk *on $n$ i.i.d. samples $\{(x_i, y_i)\}_{i=1}^n$ is*

$$\hat{R}_n(F) := \frac{1}{n} \sum_{i=1}^n \ell(F(x_i), y_i).$$

*The quantity of interest in this paper is their difference:*

$$R(F) - \hat{R}_n(F),$$

*which measures how far the empirical risk deviates from the population risk.*

**Remark.**  In practice, we compute the empirical risk using the training set, and approximate the population risk using the test set.

**Definition 9** (Bayes Predictor). *The* Bayes predictor *is the ideal predictor that has the same model architecture as $F$ but is trained with full access to the true population distribution. Formally, it is the conditional risk minimizer*

$$F_k^*(z) := \arg\min_f \; \mathbb{E}[\ell(f(z), Y) \mid Z_k = z].$$

*Intuitively, $F_k^*$ returns the best possible prediction given the information contained in $Z_k$. Moreover, since $F_k^*$ is optimized with respect to the full population distribution, we assume that its loss on any sample $(z, y)$ is uniformly bounded by some finite constant $C_\ell < \infty$.*

**Remark.**  The Bayes predictor serves as an ideal reference model in our analysis, returning the population-optimal output that minimizes the conditional risk given the embedding $Z_k$. Replacing the discrete label $Y$ with the continuous output $F_k^*(Z_k)$ effectively *smooths the labels* and ensures that the loss becomes differentiable with respect to the embeddings. However, since $F_k^*(Z_k)$ does not exactly match the true label $Y$, this substitution introduces an *approximation error*, whose effect must be explicitly controlled in our theoretical bounds.

### 3.5 STANDING ASSUMPTIONS

We impose the following regularity assumptions, stated with explicit constants to clarify their roles in later bounds.

**Assumption 1** (Measurability of embeddings). *For each layer $k$, the embedding map $h_{\leq k} : \mathcal{X} \to \mathcal{Z}_k$ is measurable, so that the pushforward distribution $\tilde{P}_k^Z$ is well defined.*

**Assumption 2** (Bounded support)**.** *Each embedding distribution $\tilde{P}_k^Z$ has bounded $\ell_1$-diameter:*

$$D_k := \sup_{z,z' \in \operatorname{supp}(\tilde{P}_k^Z)} \|z - z'\|_1 < \infty.$$

*The bounded diameter $D_k$ is used in Proposition 2 (Appendix A.4.4) to control the effect of a single-sample replacement when applying McDiarmid's inequality to the Wasserstein term $W_1(\tilde{P}_k^Z, \hat{P}_{k,n}^Z)$.*

**Assumption 3** (Lipschitz continuity of tail and Bayes maps)**.** *For each layer $k$, consider an open neighborhood $U_k \supseteq \operatorname{supp}(\tilde{P}_k^Z)$ of the embedding support. The network tail map $F$ and the Bayes predictor $F_k^*$ are assumed Lipschitz on $U_k$, with constants*

$$L_k(F) := \sup_{z \in U_k} \|\nabla F(z)\|_{\operatorname{op}}, \qquad L_k(F^*) := \sup_{z \in U_k} \|\nabla F_k^*(z)\|_{\operatorname{op}}.$$

*These Lipschitz constants are used in Lemma 2 to bound the gradient of the layerwise loss $g_k(z) = \ell(F(z), F_k^*(z))$, yielding $L_k(g) \le L_k(F)M_F + L_k(F^*)M_{F^*}$.*

**Assumption 4** (Loss regularity)**.** *The loss $\ell : \mathbb{R}^C \times \mathbb{R}^C \to \mathbb{R}$ is continuously differentiable in both arguments. There exist constants $M_F, M_{F^*} < \infty$ such that*

$$\|\nabla_u \ell(u,v)\|_\infty \le M_F, \qquad \|\nabla_v \ell(u,v)\|_\infty \le M_{F^*}.$$

*The gradient bounds $M_F$ and $M_{F^*}$ are used in Lemma 2 through the chain rule to obtain the Lipschitz constant of $g_k$, and in Lemma 3 to control the approximation error incurred when replacing labels by the Bayes predictor, i.e., the term involving $\ell(F(x), F_k^*(x)) - \ell(F(x), y)$.*

# 4 MAIN THEORETICAL RESULTS

## 4.1 DIMENSION-DEPENDENT GENERALIZATION BOUND

Our first main result shows that the generalization error can be controlled in terms of the intrinsic dimension of intermediate embeddings, together with Lipschitz sensitivity factors.

**Theorem 4.1** (Dimension-dependent generalization bound)**.** *Assume Assumptions 1–4. Fix confidence $\delta \in (0,1)$. Suppose that for each layer $k$ there exist constants $C_k > 0$, intrinsic dimension $d_k > 0$, and arbitrarily small $\epsilon > 0$ such that, for all sufficiently large $n$,*

$$\mathbb{E}\big[\mathcal{W}_1(\tilde{P}_k^Z, \hat{P}_{k,n}^Z)\big] \le C_k\, n^{-1/(d_k+\epsilon)}.$$

*Then for any fixed predictor $F \in \mathcal{F}$, with probability at least $1 - \delta$,*

$$
\begin{aligned}
R(F) \le \hat{R}_n(F) + \min_{0 \le k \le L} \Big\{ & \bar{L}_k\Big(\mathbb{E}[\mathcal{W}_1(\tilde{P}_k^Z, \hat{P}_{k,n}^Z)] + D_k\sqrt{\tfrac{1}{2n}\log\tfrac{2(L+1)}{\delta}}\Big) \\
& + M_{F^*}\Big(2\,\mathbb{E}\|Y - F_k^*(Z)\|_1 + \sqrt{\tfrac{2}{n}\log\tfrac{2(L+1)}{\delta}}\Big)\Big\} \\
\le \hat{R}_n(F) + \min_{0 \le k \le L} \Big\{ & \bar{L}_k\Big(C_k\, n^{-1/(d_k+\epsilon)} + D_k\sqrt{\tfrac{1}{2n}\log\tfrac{2(L+1)}{\delta}}\Big) \\
& + M_{F^*}\Big(2\,\mathbb{E}\|Y - F_k^*(Z)\|_1 + \sqrt{\tfrac{2}{n}\log\tfrac{2(L+1)}{\delta}}\Big)\Big\}. \quad (1)
\end{aligned}
$$

*where $D_k$ is the $\ell_1$-diameter of the embedding support, the intrinsic dimension $d_k$ is precisely the upper Wasserstein dimension of the embedding distribution $\tilde{P}_k^Z$, and*

$$\bar{L}_k := L_k(F)\, M_F + L_k(F^*)\, M_{F^*}.$$

**Remarks.**

- **Dimension dependence.** The term $n^{-1/(d_k+\epsilon)}$ is the dominant statistical rate. It comes from how quickly the empirical embedding distribution converges to its population counterpart. If the embeddings at layer $k$ concentrate on a low-dimensional set (small $d_k$), the Wasserstein distance shrinks faster, so fewer samples are needed to approximate the true embedding distribution. This explains why models that compress information into lower-dimensional representations tend to generalize better.

- **Sensitivity.** The factor $\bar{L}_k$ quantifies how sensitive the loss is to perturbations in the embeddings. It combines the Lipschitz constant of the model's tail ($L_k(F)$) and that of the Bayes predictor ($L_k(F^*)$), scaled by the loss derivative bounds. Intuitively, even if embeddings concentrate in a low-dimensional region, the benefit may be offset if the predictor reacts too strongly to small embedding changes. Thus $\bar{L}_k$ captures the architectural and task-dependent smoothness required for low dimension to translate into good generalization.
- **Layer minimization.** Each layer provides a different balance between dimension and sensitivity. Early layers may have higher intrinsic dimension but lower sensitivity, while later layers may be more compressed but more sensitive. The bound holds for all layers, so taking the minimum over $k$ automatically selects the representation whose dimensionsensitivity tradeoff gives the tightest control of the generalization gap.

## 4.2 FINAL-LAYER SIMPLIFICATION

When we analyze the embeddings at the output layer, the expression simplifies further. At this layer, the downstream network mapping is the identity: the output of the network is exactly the embedding $Z_L$. Therefore the architectural Lipschitz constant disappears, i.e. $L_L(F) = 1$.

**Corollary 1** (Final-layer bound). *For the final embedding $Z_L = F(X)$ we have $L_L(F) = 1$. With probability at least $1 - \delta$,*

$$R(F) \leq \hat{R}_n(F) + \left(M_F + L_L(F^*)\,M_{F^*}\right)\left(C_L\,n^{-1/(d_L+\epsilon)} + D_L\sqrt{\tfrac{1}{2n}\log\tfrac{2(L+1)}{\delta}}\right) \tag{2}$$

$$+ M_{F^*}\left(2\,\mathbb{E}\|Y - F_L^*(Z_L)\|_1 + \sqrt{\tfrac{2}{n}\log\tfrac{2(L+1)}{\delta}}\right).$$

**Remark.** At the final layer, the architectural Lipschitz factors vanish, leaving a bound that depends only on: (i) the intrinsic dimension $d_L$, (ii) the embedding diameter $D_L$, (iii) the loss-derivative constants ($M_F, M_{F^*}$), and (iv) the Bayes smoothness constant $L_L(F^*)$ together with the irreducible label-noise term. This shows that final-layer embeddings provide a particularly convenient diagnostic: generalization is largely driven by dimension and data-related smoothness, without additional sensitivity to the network's architectural Lipschitz constant.

**Summary.** The generalization error is determined by two main forces: the intrinsic dimension of embeddings (statistical efficiency) and Lipschitz sensitivity (stability to perturbations). Intermediate layers reflect both effects, requiring joint consideration of dimension and sensitivity. The final layer provides a simplified diagnostic where only dimension and distribution-dependent smoothness remain, clarifying why final-layer dimension has strong predictive power for generalization. The complete proof of Theorem 4.1 is provided in Appendix A.

## 5 EXPERIMENTS AND RESULTS

### 5.1 VALIDATION OF WASSERSTEIN CONVERGENCE SCALING

Theorem 3.1 shows that the convergence rate of empirical to population distributions in Wasserstein distance is governed by the intrinsic dimension. A key question is whether this scaling law also holds for the complex embeddings produced by neural networks. To validate this, we train a five-layer MLP autoencoder on MNIST and analyze how the Wasserstein distance between empirical and population embeddings depends on both intrinsic dimension and sample size.

We examine two perspectives. First, we fix the sample size $n$ and evaluate how Wasserstein distance varies with intrinsic dimension. Second, we fix intrinsic dimension and study how the Wasserstein distance decreases with $n$ according to the predicted $n^{-1/(d+\epsilon)}$ law. In the experiments, we vary sample sizes from 100 to 1500 and compute embeddings from the trained autoencoder. For each configuration, we estimate the intrinsic dimension of embeddings by MLE (Levina & Bickel, 2004) and compute the Wasserstein distance between two embedding sets of size $n$: one drawn as the empirical sample set, and another drawn independently to approximate the population embedding distribution. This empirical Wasserstein distance quantifies how closely the finite sample set approximates the broader embedding distribution under the chosen metric.

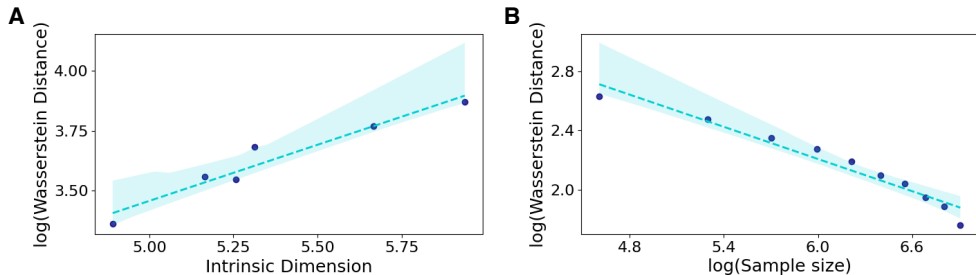

Figure 2: **Scaling of Wasserstein Convergence in Neural Network Embeddings. (A)** With fixed sample size, log(Wasserstein distance) increases approximately linearly with embedding dimension. **(B)** With fixed embedding dimension, log(Wasserstein distance) decreases approximately linearly with log(sample size).

The results reveal two consistent patterns. For fixed $n$, the log of Wasserstein distance increases approximately linearly with intrinsic dimension, consistent with the exponential dependence on $d$ predicted by the theory (Figure 2A). For fixed $d$, the log of Wasserstein distance decreases approximately linearly with $\log n$, as predicted by the $n^{-1/(d+\epsilon)}$ law (Figure 2B). These findings validate that the relationship between intrinsic dimension, sample size, and Wasserstein convergence extends to real neural network embeddings. Other details are provided in Appendix B.1.

## 5.2 RELATIONSHIP BETWEEN INTRINSIC DIMENSION, WASSERSTEIN DISTANCE AND GENERALIZATION GAP

Corollary 1 shows that when analyzing the final layer's embedding, architectural sensitivity vanishes, and the generalization gap $R(F) - \hat{R}_n(F)$ is largely governed by the intrinsic dimension of the embedding. To test this prediction in realistic settings, we evaluate ResNet-18, 34, 50, 101 and 152 on CIFAR-10 and CIFAR-100.

For each trained model, we extract the final-layer embeddings and estimate their intrinsic dimension $d_L$. We also compute the empirical Wasserstein distance $\mathcal{W}_1$ between validation and test embedding distributions. To obtain finer-grained insight, we perform the analysis at the class level rather than only at the aggregate level: each ResNet model yields 10 data points on CIFAR-10 (one per class) and 100 data points on CIFAR-100. This allows us to assess the relationship between embedding properties and the generalization gap more accurately.

Figure 3 shows that both the intrinsic dimension $d_L$ of the final embedding and the empirical Wasserstein distance $\mathcal{W}_1$ correlate positively with the observed generalization gap across architectures and datasets, consistent with the relationship predicted by our generalization error bound.

These experiments extend our earlier results from MNIST to more complicated datasets and architectures. Overall, the results reinforce the central theoretical insight: at the final layer, architectural factors vanish and lower intrinsic dimension is strongly associated with smaller generalization gaps. Other details are provided in Appendix B.2. We also compare different hyperparameter choices and estimation algorithms in Appendix C, all of which yield results consistent with those reported here.

## 5.3 INTERVENTIONS ON NETWORK WIDTH

Theorem 4.1 emphasizes that when analyzing intermediate-layer embeddings, the generalization error depends jointly on the embedding dimension and the Lipschitz constant of the downstream mapping from embedding to output. To empirically investigate this relationship, we conduct controlled interventions by varying the width of a middle layer in a neural network.

Analyzing the Lipschitz constant of a general neural network is challenging. To facilitate this analysis, we consider fully connected MLPs with ReLU activations, for which the product of the spectral norms of the weight matrices provides an upper bound on the network's Lipschitz constant (Bartlett et al., 2017). We use this bound as a proxy for the network's Lipschitz constant and systematically

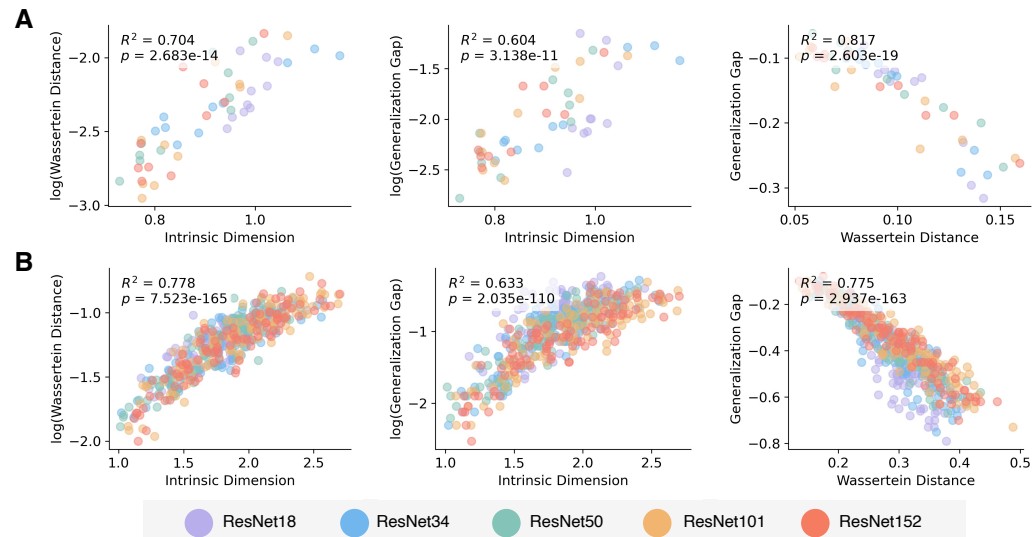

Figure 3: **Relationship Between Final-Layer Embedding Dimension, Wasserstein Distance and Generalization Error.** We evaluate CIFAR-10 (**A**) and CIFAR-100 (**B**) and observe a significant correlation between final-layer embedding dimension, Wasserstein distance and generalization error. This pattern aligns with predictions from the generalization error bound, indicating the bound is sufficiently tight and that embedding dimension, together with Wasserstein convergence, provides an effective predictor of generalization error.

study how changes in the width of a single intermediate layer affect the intrinsic dimension of the embedding, the network's Lipschitz constant and the resulting generalization error.

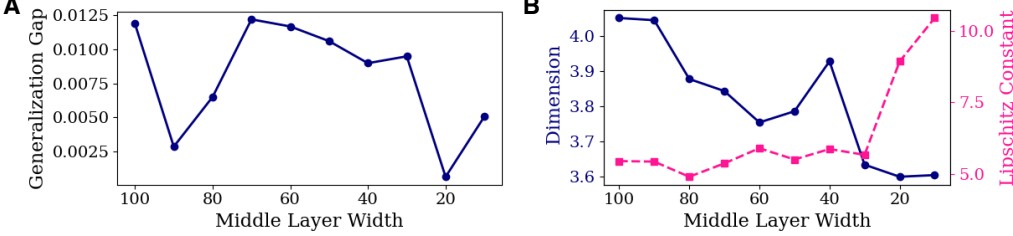

Figure 4: **Effect of Network Width on Intrinsic Dimension and Generalization Error.** (**A**) A six-layer MLP is used to vary the width of the third layer. Reducing width does not necessarily decrease generalization error. (**B**) Narrower networks reduce embedding intrinsic dimension, but the network Lipschitz constant increases, offsetting the benefit of lower dimension. Hence, both embedding dimension and Lipschitz constant of network should be jointly considered when analyzing generalization error.

Specifically, we train a six-layer MLP on CIFAR-10 and vary the width of the third layer from 100 down to 10. Figure 4 summarizes the results: Panel A shows that as the layer width decreases, the generalization error does not consistently decline. Panel B shows that the embedding's intrinsic dimension steadily decreases with narrower layers, while the network's Lipschitz constant increases, particularly when the width drops below 30. This increase in sensitivity likely offsets the benefit of lower intrinsic dimension, explaining why the generalization error does not significantly improve.

These results confirm that simply reducing network width does not reliably enhance generalization. Narrower layers can reduce the embedding dimension, but this effect may be counteracted by increased sensitivity of the downstream mapping. The findings provide empirical support for Theorem 4.1 and highlight the importance of jointly considering embedding geometry and Lipschitz

sensitivity when analyzing overparameterized networks. Other details are provided in Appendix B.3.

## 6 DISCUSSION AND CONCLUSION

Understanding why deep networks generalize despite massive overparameterization remains a central challenge. This work advances a *representation-centric view*, showing that generalization error can be related to two measurable properties: the intrinsic dimension $d_k$ of embeddings and a sensitivity term $\bar{L}_k$ that captures how embedding perturbations propagate through the network. These quantities integrate model structure and data distribution, offering post-hoc diagnostics beyond classical capacity-based bounds. Experiments across architectures and datasets confirm this interplay, showing that embedding dimension, Wasserstein distance, and generalization error track each other consistently, with scaling close to $n^{-1/(d+\epsilon)}$, and that architectural sensitivity vanishes at the final layer so dimension plays a dominant role.

In Appendix C.5, we further extend these analyses to large models and ImageNet, demonstrating that the correlations among dimension, Wasserstein distance and generalization error persist even at large-scale model and dataset. Additionally, we analyze layer-wise embeddings in ResNet-154 in Appendix C.6, finding that while dimension and Wasserstein distance remain strongly correlated across layers, the correlations between dimension and generalization performance, as well as between Wasserstein distance and generalization performance, increase progressively with network depth. These results reinforce the importance of embedding geometry in explaining generalization behavior, particularly in deeper layers and larger-scale settings.

**Limitations.** Our bound contains constants that may be loose. However, our experiments demonstrate a significant correlation between embedding dimension, Wasserstein distance and generalization error, indicating that changes in generalization error can be effectively captured by variations in dimension. This empirical alignment shows that our bound, despite potentially loose constants, retains practical significance as a diagnostic tool for generalization. We also rely on assumptions such as the Lipschitz continuity of the Bayes predictor, which ensures a well-defined and bounded relationship between inputs and outputs. Without such assumptions, it is not possible to derive a generalization bound purely from properties of a specific layer's embedding. Relaxing these assumptions is an important direction for future work. Estimating the sensitivity term $\bar{L}_k$ remains challenging in practice, and developing reliable estimators is necessary for broader applicability. Finally, in our current analysis we treat embedding dimension and network sensitivity as independent. In reality, these quantities may be correlated. Understanding this interplay is an important direction for future work.

**Conclusion.** By shifting focus from network parameter to embedding geometry, we identify intrinsic dimension and sensitivity as core drivers of generalization. This framework offers both a theoretical foundation and practical tools for analyzing and designing deep networks.

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

# A SUPPLEMENT OF THEORETICAL RESULTS

Before presenting the detailed proofs, we first summarize the key notation used throughout this appendix and the main paper. This notation table serves as a convenient reference to improve clarity and readability.

**Notation summary.** Key symbols used throughout the paper.

Table 1: Notation summary for key symbols in the paper.

| Symbol | Meaning |
|--------|---------|
| $\tilde{P}_k^Z$ | Population embedding distribution at layer $k$. |
| $\hat{\tilde{P}}_{k,n}^Z$ | Empirical embedding distribution from $n$ samples. |
| $\hat{\mu}_n$ | Empirical measure of $n$ i.i.d. samples. |
| $R(F)$ | Population risk of predictor $F$. |
| $\hat{R}_n(F)$ | Empirical risk on validation set. |
| $\mathrm{gen}(F)$ | Generalization gap $R(F) - \hat{R}_n(F)$. |
| $F_{\leq k}$ | Encoder mapping input $x$ to embedding $z$ at layer $k$. |
| $F_k$ | Tail map from layer-$k$ embedding $z$ to output. |
| $F(x)$ | Overall predictor $F_k(F_{\leq k}(x))$. |
| $F_k^*$ | Bayes predictor from layer-$k$ embedding $z$ to output. |
| $d_k$ | Intrinsic dimension of $\tilde{P}_k^Z$. |
| $D_k$ | $\ell_1$-diameter of support of $\tilde{P}_k^Z$. |
| $\mathcal{W}_1(\cdot, \cdot)$ | 1-Wasserstein distance. |
| $L_k(F)$ | Lipschitz constant of network tail from layer $k$ to output. |
| $L_k(F^*)$ | Lipschitz constant of Bayes predictor from layer $k$ to output. |
| $M_F$ | Bound on loss gradient wrt network output. |
| $M_{F^*}$ | Bound on loss gradient wrt Bayes predictor output. |
| $\ell$ | Loss function (e.g., squared loss, cross-entropy). |
| $B_\ell$ | Uniform bound on loss values. |

**Roadmap of the appendix.** This appendix provides a complete derivation of the dimension-dependent generalization bound stated in Theorem 4.1. The proof is organized into four self-contained steps:

1. **Preliminaries (Subsection A.1):** We collect standard technical tools used throughout the proofs, including optimal transport results and Wasserstein bounds for Lipschitz functions.
2. **Risk decomposition via Bayes surrogates (Subsection A.2):** In classification settings, labels are discrete, so the observed loss is non-differentiable with respect to embeddings. We introduce layer-wise Bayes predictors as continuous surrogates, leading to a decomposition of the generalization error into three terms: (A) approximation gap, (B) oracle statistical gap, and (C) empirical model gap.
3. **Controlling the decomposed terms (Subsection A.4):** Each term is bounded explicitly. (A) and (C) are controlled by irreducible label noise, while (B) is controlled via the 1-Wasserstein distance between empirical and population embeddings combined with the oracle loss Lipschitz constant. Concentration inequalities yield high-probability bounds scaling with embedding dimension and sample size.
4. **Recovering network effects (Subsection A.5):** The oracle Lipschitz constant $L_k(g)$ is decomposed as

$$L_k(g) \leq L_k(F) \, M_F + L_k(F^*) \, M_{F^*},$$

separating controllable network-dependent and intrinsic Bayes predictor contributions. Substituting this into the previous bounds connects embedding geometry, statistical concentration, and network design.

Overall, these steps provide a clear, high-probability generalization bound that disentangles statistical, architectural, and label-noise contributions.

## A.1 Preliminaries: Useful Lemmas and Theorems

In this subsection we collect several standard results that will be used throughout the proofs. They are presented here to avoid interruptions in the main arguments later.

### A.1.1 Existence of Optimal Transport Plan

**Theorem A.1** (Existence of Optimal Transport Plan (Villani, 2009, Theorem 4.1)). *Let $(\mathcal{X}, \mu)$ and $(\mathcal{Y}, \nu)$ be Polish probability spaces, and let*

$$c : \mathcal{X} \times \mathcal{Y} \longrightarrow \mathbb{R} \cup \{+\infty\}$$

*be a lower semicontinuous cost function. Then there exists a coupling $\gamma^* \in \Pi(\mu, \nu)$ that minimizes the expected cost:*

$$\int_{\mathcal{X} \times \mathcal{Y}} c(x, y) \, d\gamma^*(x, y) = \inf_{\gamma \in \Pi(\mu,\nu)} \int_{\mathcal{X} \times \mathcal{Y}} c(x, y) \, d\gamma(x, y).$$

*In particular, for the 1-Wasserstein cost $c(z, z') = \|z - z'\|_1$ on a Polish space there exists an optimal coupling attaining $\mathcal{W}_1$.*

### A.1.2 Wasserstein Bound for Lipschitz Functions

**Lemma 1** (Expectation difference controlled by $W_1$). *Let $\mu, \nu$ be probability measures on $\mathbb{R}^d$. If $h : \mathbb{R}^d \to \mathbb{R}$ is $L_h$-Lipschitz with respect to $\ell_1$ (i.e. $|h(z) - h(z')| \leq L_h \|z - z'\|_1$ for all $z, z'$), then*

$$\left| \int h \, d\mu - \int h \, d\nu \right| \leq L_h \, \mathcal{W}_1(\mu, \nu).$$

*Proof.* By definition of $\mathcal{W}_1$ and for any coupling $\pi \in \Pi(\mu, \nu)$,

$$\int h \, d\mu - \int h \, d\nu = \iint \big( h(z) - h(z') \big) \, d\pi(z, z') \leq \iint L_h \|z - z'\|_1 \, d\pi(z, z').$$

Taking infimum over all couplings $\pi$ gives the claim. The absolute value follows by symmetry (swapping $\mu, \nu$). □

## A.2 Risk Decomposition via Bayes Surrogates

**Motivation.** In classification, the label $Y$ is discrete, so the observed loss $\ell(F(X), Y)$ is not differentiable with respect to embeddings $Z_k$. This obstructs a direct Lipschitz-based analysis of the risk, which is central to our approach. To address this, we introduce at each layer $k$ the *Bayes predictor* $F_k^*(Z_k)$, a continuous surrogate for the discrete label. Replacing $Y$ with $F_k^*(Z_k)$ yields the *oracle loss*, which is differentiable in $Z_k$ and hence amenable to Lipschitz/Wasserstein analysis. The cost of this replacement is an additional error term capturing the mismatch between observed and oracle risks. This term corresponds to irreducible label randomness and will be explicitly controlled.

**Definition 10** (Observed and oracle risks). *Let $\ell : \mathbb{R}^C \times \mathbb{R}^C \to \mathbb{R}$ be a measurable loss. At the input layer, the observed risks are*

$$R_0^{\text{obs}} := \mathbb{E}_{(X,Y) \sim \mathcal{D}}[\ell(F(X), Y)], \qquad \hat{R}_{0,n}^{\text{obs}} := \frac{1}{n} \sum_{i=1}^n \ell(F(x_i), y_i).$$

*At layer $k$, the oracle loss is defined as*

$$g_k(z) := \ell(F_k(z), F_k^*(z)),$$

*with population and empirical oracle risks*

$$R_k^{\text{oracle}} := \mathbb{E}_{Z_k \sim \tilde{P}_k^Z}[g_k(Z_k)], \qquad \hat{R}_{k,n}^{\text{oracle}} := \frac{1}{n} \sum_{i=1}^n g_k(z_{k,i}).$$

**Proposition 1** (Risk decomposition)**.** *For any predictor $F$ and any intermediate layer $k$,*

$$R_0^{\text{obs}} - \hat{R}_{0,n}^{\text{obs}} = (R_0^{\text{obs}} - R_k^{\text{oracle}}) + (R_k^{\text{oracle}} - \hat{R}_{k,n}^{\text{oracle}}) + (\hat{R}_{k,n}^{\text{oracle}} - \hat{R}_{0,n}^{\text{obs}}).$$

**Interpretation.** The decomposition separates the generalization gap into three terms:

- **Approximation gap:** $R_0^{\text{obs}} - R_k^{\text{oracle}}$ measures the loss of information when replacing discrete labels by the Bayes predictor at layer $k$.
- **Oracle statistical gap:** $R_k^{\text{oracle}} - \hat{R}_{k,n}^{\text{oracle}}$ is the population-to-sample deviation of the oracle loss, the term to be controlled via Lipschitz continuity and Wasserstein concentration.
- **Empirical model gap:** $\hat{R}_{k,n}^{\text{oracle}} - \hat{R}_{0,n}^{\text{obs}}$ quantifies how network predictions differ from Bayes-optimal predictions under the empirical distribution.

**Summary.** The observed generalization error is thus expressed as an oracle component (amenable to Lipschitz/Wasserstein analysis) plus two additional error terms that capture irreducible label noise and model approximation. This motivates analyzing the Lipschitz constant of the oracle loss $g_k(z)$, which we do next.

## A.3 Lipschitz Constant of the Layer-Wise Loss

Having introduced the oracle loss $g_k(z) = \ell(F_k(z), F_k^*(z))$, we now analyze its Lipschitz continuity with respect to the embedding $z$. This is possible because both arguments of $g_k$ are continuous functions of $z$.

**Gradient and Lipschitz bound.**

**Lemma 2.** *Suppose Assumptions 1–4 hold. Then for all $z \in U_k$,*

$$\nabla g_k(z) = \nabla F_k(z)^\top \, \partial_F \ell(F_k(z), F_k^*(z)) + \nabla F_k^*(z)^\top \, \partial_{F^*} \ell(F_k(z), F_k^*(z)).$$

*Hence*

$$\|\nabla g_k(z)\|_\infty \leq \|\nabla F_k(z)\|_{\text{op}} \|\partial_F \ell\|_\infty + \|\nabla F_k^*(z)\|_{\text{op}} \|\partial_{F^*} \ell\|_\infty.$$

*If $\|\partial_F \ell\|_\infty \leq M_F$, $\|\partial_{F^*} \ell\|_\infty \leq M_{F^*}$, and the Jacobians satisfy $\|\nabla F_k(z)\|_{\text{op}} \leq L_k(F)$, $\|\nabla F_k^*(z)\|_{\text{op}} \leq L_k(F^*)$, then*

$$L_k(g) := \sup_{z \in U_k} \|\nabla g_k(z)\|_\infty \leq L_k(F)\, M_F + L_k(F^*)\, M_{F^*}.$$

*Proof.* The chain rule gives the gradient expression. Applying $\|A^\top v\|_\infty \leq \|A\|_{\text{op}} \|v\|_\infty$ and substituting the uniform bounds yields the claim. $\qquad\qquad\square$

**Remark.** The bound cleanly separates two contributions: (i) the network-dependent Lipschitz constant $L_k(F)$, controllable by architecture or regularization, and (ii) the distribution-dependent Lipschitz constant $L_k(F^*)$, reflecting the inherent complexity of the Bayes predictor. Thus the oracle loss Lipschitz constant factors into a controllable and an uncontrollable component, which will play distinct roles in the final generalization bound.

## A.4 Controlling the Decomposed Terms

**Overview of the approach.** Proposition 1 decomposes the generalization gap into three terms:

$$\underbrace{R_0^{\text{obs}} - R_k^{\text{oracle}}}_{(A)\text{ approximation gap}}, \quad \underbrace{R_k^{\text{oracle}} - \hat{R}_{k,n}^{\text{oracle}}}_{(B)\text{ oracle statistical gap}}, \quad \underbrace{\hat{R}_{k,n}^{\text{oracle}} - \hat{R}_{0,n}^{\text{obs}}}_{(C)\text{ empirical model gap}}.$$

We now control these terms separately:

- (A) measures the error incurred by replacing labels $Y$ with the Bayes surrogate $F_k^*(Z_k)$.
- (B) measures the statistical deviation between population and empirical distributions of embeddings, for the oracle loss.
- (C) measures the discrepancy between network predictions and Bayes-optimal predictions under the empirical distribution.

Each of (A), (B), (C) will be treated in turn.

### A.4.1 BOUNDING THE APPROXIMATION GAP (A).

**Lemma 3** (Control of approximation gap). *Assume the loss $\ell : \mathbb{R}^C \times \mathbb{R}^C \to \mathbb{R}$ is Lipschitz in its second argument with constant $M_{F^*}$ (Assumption 4). Then for any predictor $F$ and any layer $k$,*

$$\left| R_0^{\mathrm{obs}} - R_k^{\mathrm{oracle}} \right| \; \leq \; M_{F^*} \, \mathbb{E}_{Z \sim \tilde{P}_k^Z} \left[ \| Y - F_k^*(Z) \|_1 \right].$$

*Proof.* For any sample $(x, y)$ with embedding $z = h_{\leq k}(x)$,

$$\left| \ell(F(x), y) - \ell(F_k(z), F_k^*(z)) \right| \leq M_{F^*} \| y - F_k^*(z) \|_1,$$

by Lipschitz continuity of $\ell$ in the second argument. Taking expectation over $(X, Y) \sim \mathcal{D}$ yields the result. $\qquad\square$

### A.4.2 BOUNDING THE ORACLE STATISTICAL GAP (B).

**Lemma 4** (Oracle risk controlled by $W_1$). *For any predictor $F \in \mathcal{F}$ and any layer $k$,*

$$R_k^{\mathrm{oracle}} - \hat{R}_{k,n}^{\mathrm{oracle}} \; \leq \; L_k(g) \, \mathcal{W}_1\!\left( \tilde{P}_k^Z, \, \hat{\tilde{P}}_{k,n}^Z \right),$$

*where $L_k(g)$ is the Lipschitz constant of $g_k(z) = \ell(F_k(z), F_k^*(z))$ with respect to the $\ell_1$-metric, as given in Lemma 2.*

*Proof.* By Kantorovich-Rubinstein duality, for any $L$-Lipschitz function $f$,

$$\left| \int f \, d\mu - \int f \, d\nu \right| \; \leq \; L \, W_1(\mu, \nu).$$

Applying this with $f = g_k$, $\mu = \tilde{P}_k^Z$, and $\nu = \hat{\tilde{P}}_{k,n}^Z$, and recalling that $g_k$ has Lipschitz constant $L_k(g)$, gives the desired bound. $\qquad\square$

### A.4.3 BOUNDING THE EMPIRICAL MODEL GAP (C).

**Lemma 5** (Control of empirical model gap). *Under the same assumptions as Lemma 3, for any predictor $F$ and any layer $k$,*

$$\left| \hat{R}_{k,n}^{\mathrm{oracle}} - \hat{R}_{0,n}^{\mathrm{obs}} \right| \; \leq \; M_{F^*} \, \frac{1}{n} \sum_{i=1}^{n} \| y_i - F_k^*(z_{k,i}) \|_1.$$

*Proof.* For each validation sample $(x_i, y_i)$ with embedding $z_{k,i} = h_{\leq k}(x_i)$,

$$\left| \ell(F(x_i), y_i) - \ell(F_k(z_{k,i}), F_k^*(z_{k,i})) \right| \leq M_{F^*} \| y_i - F_k^*(z_{k,i}) \|_1.$$

Averaging over $i = 1, \ldots, n$ yields the result. $\qquad\square$

### A.4.4 CONCENTRATION OF $T_k := \mathcal{W}_1(\tilde{P}_k^Z, \hat{\tilde{P}}_{k,n}^Z)$ AND OF THE EMPIRICAL NOISE AVERAGE

**Motivation.** The deterministic decomposition in Proposition 1 reduces the generalization gap to three terms. Among them, two depend explicitly on the randomness of the empirical sample:

- the Wasserstein distance $T_k = \mathcal{W}_1(\tilde{P}_k^Z, \hat{\tilde{P}}_{k,n}^Z)$, which controls the oracle statistical gap (B);
- the empirical noise average $\bar{u}^{(k)} = \frac{1}{n} \sum_{i=1}^{n} \| y_i - F_k^*(z_{k,i}) \|_1$, which appears in the empirical model gap (C).

To obtain a high-probability generalization bound, it is therefore crucial to quantify how much these quantities deviate from their expectations. We now prove two concentration inequalities: a bounded-difference bound (McDiarmid) for $T_k$, and a Hoeffding bound for $\bar{u}^{(k)}$.

**Proposition 2** (Concentration of $T_k$ and $\bar{u}^{(k)}$). *Let $D_k := \sup_{z,z' \in \mathrm{supp}(\tilde{P}_k^Z)} \|z - z'\|_1 < \infty$ be the $\ell_1$-diameter of the embedding support. Define $T_k = \mathcal{W}_1(\tilde{P}_k^Z, \hat{\tilde{P}}_{k,n}^Z)$, and $\bar{u}^{(k)} = \frac{1}{n} \sum_{i=1}^{n} u_i^{(k)}$ with $u_i^{(k)} = \|y_i - F_k^*(z_{k,i})\|_1$. Assume $u_i^{(k)} \in [0, 2]$ for all $i$ (normalization as in the main text). Then for any $\delta \in (0, 1)$, with probability at least $1 - \frac{\delta}{2(L+1)}$,*

$$T_k \leq \mathbb{E}[T_k] + D_k \sqrt{\frac{1}{2n} \log \frac{2(L+1)}{\delta}}, \tag{3}$$

$$\bar{u}^{(k)} \leq \mathbb{E}[u^{(k)}] + \sqrt{\frac{2}{n} \log \frac{2(L+1)}{\delta}}. \tag{4}$$

*Proof.* **Step 1: Bounded-difference inequality for $T_k$.** We use the Kantorovich-Rubinstein dual representation of $W_1$:

$$\mathcal{W}_1(\mu, \nu) = \sup_{\substack{f:\mathbb{R}^d \to \mathbb{R} \\ \mathrm{Lip}(f) \leq 1}} \left\{ \int f \, d\mu - \int f \, d\nu \right\},$$

with Lipschitz constant measured in the $\ell_1$-norm. Let the empirical measure be $\hat{\tilde{P}}_{k,n}^Z = \frac{1}{n} \sum_{i=1}^{n} \delta_{z_{k,i}}$. Consider two samples $S = (z_{k,1}, \ldots, z_{k,n})$ and $S^{(j)}$ that differ only in the $j$-th element. Denote $T_k(S) = \mathcal{W}_1(\tilde{P}_k^Z, \hat{\tilde{P}}_{k,n}^Z(S))$. Then

$$|T_k(S) - T_k(S^{(j)})| \leq \frac{1}{n} \|z_{k,j} - z'_{k,j}\|_1 \leq \frac{D_k}{n}.$$

Thus $T_k$ satisfies a bounded-difference property with sensitivity $D_k/n$. Applying McDiarmid's inequality gives, for any $t > 0$,

$$\mathbb{P}(T_k - \mathbb{E}[T_k] \geq t) \leq \exp\left( -\frac{2nt^2}{D_k^2} \right).$$

Choosing $t = D_k \sqrt{\frac{1}{2n} \log \frac{2(L+1)}{\delta}}$ yields equation 3.

**Step 2: Hoeffding bound for $\bar{u}^{(k)}$.** Each $u_i^{(k)} \in [0, C_\ell]$ by Definition 9. By Hoeffding's inequality, for any $t > 0$,

$$\mathbb{P}(\bar{u}^{(k)} - \mathbb{E}[\bar{u}^{(k)}] \geq t) \leq \exp\left( -\frac{nt^2}{2} \right).$$

Choosing $t = \sqrt{\frac{2}{n} \log \frac{2(L+1)}{\delta}}$ yields equation 4.

This completes the proof. $\qquad\square$

**Discussion.** This result ensures that both the statistical fluctuation of the embedding distribution (through $T_k$) and the empirical noise magnitude (through $\bar{u}^{(k)}$) remain close to their expectations with high probability. These concentration bounds are the key probabilistic ingredients needed to convert the deterministic decomposition of the generalization gap into a high-probability generalization bound.

### A.4.5 COMBINED DETERMINISTIC AND HIGH-PROBABILITY BOUND

**Motivation.** We now combine the pieces developed above. Recall that the observed generalization gap

$$R_0^{\mathrm{obs}} - \hat{R}_{0,n}^{\mathrm{obs}}$$

was decomposed into three terms (Proposition 1). We provided deterministic bounds for each term (Lemmas 3–5), and then concentration inequalities for the random quantities $T_k$ and $\bar{u}^{(k)}$ (Proposition 2). Here we integrate these ingredients into a single high-probability generalization bound.

**Proposition 3** (High-probability control of the generalization gap)**.** *Assume Assumptions 1–4. Suppose that for each layer $k$ there exist constants $C_k > 0$, arbitrarily small $\epsilon > 0$ and $d_k > 0$ such that $\mathbb{E}[T_k] \leq C_k n^{-1/(d_k+\epsilon)}$ for all sufficiently large $n$. Fix confidence $\delta \in (0,1)$. Then with probability at least $1 - \delta$, simultaneously for all layers $k = 0, \dots, L$ and any fixed predictors $F \in \mathcal{F}$,*

$$R_0^{\mathrm{obs}} - \hat{R}_{0,n}^{\mathrm{obs}} \leq L_k(g)\Big(C_k n^{-1/(d_k+\epsilon)} + D_k \sqrt{\tfrac{1}{2n} \log \tfrac{2(L+1)}{\delta}}\Big)$$

$$+ M_{F^*}\Big(2\,\mathbb{E}\|Y - F_k^*(Z)\|_1 + \sqrt{\tfrac{2}{n} \log \tfrac{2(L+1)}{\delta}}\Big). \tag{5}$$

*Equivalently, the bound can be summarized as*

$$\boxed{R_0^{\mathrm{obs}} - \hat{R}_{0,n}^{\mathrm{obs}} \;\lesssim\; L_k(g)\, n^{-1/(d_k+\epsilon)} \;+\; M_{F^*}\,\mathbb{E}\|Y - F_k^*(Z)\|_1 \;+\; \sqrt{\tfrac{\log(L/\delta)}{n}}\,\big(L_k(g)D_k + M_{F^*}\big)}$$

*Proof.* **Step 1: Decomposition.** By Proposition 1,

$$R_0^{\mathrm{obs}} - \hat{R}_{0,n}^{\mathrm{obs}} = (A) + (B) + (C).$$

**Step 2: Deterministic bounds.** From Lemmas $3 - 5$,

$$R_0^{\mathrm{obs}} - \hat{R}_{0,n}^{\mathrm{obs}} \leq M_{F^*}\,\mathbb{E}\|Y - F_k^*(Z)\|_1 \;+\; L_k(g)\,T_k \;+\; M_{F^*}\,\bar{u}^{(k)}.$$

**Step 3: Concentration.** By Proposition 2, with probability at least $1 - \delta$,

$$T_k \leq \mathbb{E}[T_k] + D_k \sqrt{\tfrac{1}{2n} \log \tfrac{2(L+1)}{\delta}}, \qquad \bar{u}^{(k)} \leq \mathbb{E}[u^{(k)}] + \sqrt{\tfrac{2}{n} \log \tfrac{2(L+1)}{\delta}}.$$

Since $\mathbb{E}[u^{(k)}] = \mathbb{E}\|Y - F_k^*(Z)\|_1$, substituting yields

$$R_0^{\mathrm{obs}} - \hat{R}_{0,n}^{\mathrm{obs}} \leq L_k(g)\Big(\mathbb{E}[T_k] + D_k \sqrt{\tfrac{1}{2n} \log \tfrac{2(L+1)}{\delta}}\Big)$$

$$+ M_{F^*}\Big(2\mathbb{E}\|Y - F_k^*(Z)\|_1 + \sqrt{\tfrac{2}{n} \log \tfrac{2(L+1)}{\delta}}\Big).$$

Finally substitute $\mathbb{E}[T_k] \leq C_k n^{-1/(d_k+\epsilon)}$ to obtain equation 5. $\qquad\square$

**Discussion.** This bound highlights three components:

- The *statistical rate* $L_k(g)\,C_k n^{-1/(d_k+\epsilon)}$ combines embedding geometry (via $d_k$) and oracle loss sensitivity (via $L_k(g)$).
- The *noise/approximation terms* $M_{F^*}\,\mathbb{E}\|Y - F_k^*(Z)\|_1$ arise from replacing discrete labels by the Bayes predictor.
- The *concentration terms* scale as $O(\sqrt{\tfrac{\log(L/\delta)}{n}})$, with constants depending on both distributional ($M_{F^*}$) and geometric ($D_k$) quantities.

Together, these yield an explicit and interpretable high-probability upper bound on the observed generalization gap.

## A.5 RECOVERING NETWORK EFFECTS VIA LIPSCHITZ CONSTANTS

In the previous subsection, the oracle statistical gap (B) was controlled using the Lipschitz constant $L_k(g)$ of the oracle loss. We now expand it to expose how the bound depends both on the network architecture (controllable) and on the data distribution (intrinsic).

### A.5.1 EXPANSION OF $L_k(g)$

From Lemma 2,

$$L_k(g) := \sup_{z \in U_k} \|\nabla g_k(z)\|_\infty \;\leq\; L_k(F)\,M_F + L_k(F^*)\,M_{F^*},$$

where:

- $L_k(F)$ is the Lipschitz constant of the tail sub-network from layer $k$ to the output;
- $L_k(F^*)$ is the Lipschitz constant of the Bayes predictor at layer $k$;
- $M_F, M_{F^*}$ are uniform derivative bounds of the loss with respect to its two arguments.

**Proof sketch.** By the chain rule,

$$\nabla g_k(z) = \nabla F_k(z)^\top \, \partial_F \ell(F_k(z), F_k^*(z)) + \nabla F_k^*(z)^\top \, \partial_{F^*} \ell(F_k(z), F_k^*(z)).$$

Applying the operator norm inequality and the uniform derivative bounds yields the stated inequality.

### A.5.2 Controllable vs. intrinsic contributions

This decomposition separates the two sources of sensitivity:

- **Network-dependent term:** $L_k(F) \, M_F$, which is determined by the architecture and training of the tail network. It can be reduced by explicit design choices (e.g., normalization layers, spectral norm constraints, Lipschitz regularization).
- **Distribution-dependent term:** $L_k(F^*) \, M_{F^*}$, which reflects the smoothness of the Bayes predictor relative to embeddings. This term is intrinsic to the data distribution and cannot be improved by network design.

### A.5.3 Implication for the generalization bound

Substituting the decomposition of $L_k(g)$ into Proposition 3 gives

$$R_0^{\mathrm{obs}} - \hat{R}_{0,n}^{\mathrm{obs}} \leq \left( L_k(F) \, M_F + L_k(F^*) \, M_{F^*} \right) \left( C_k n^{-1/(d_k+\epsilon)} + D_k \sqrt{\frac{1}{2n} \log \frac{2(L+1)}{\delta}} \right)$$
$$+ \underbrace{\left[ M_{F^*} \left( 2\mathbb{E}\|Y - F_k^*(Z)\|_1 + \sqrt{\frac{2}{n} \log \frac{L}{\delta}} \right) \right]}_{\text{Bayes surrogate terms}}. \tag{6}$$

Thus the final bound reflects two complementary mechanisms:

1. *Embedding geometry:* the intrinsic dimension $s_k$ governs the statistical rate of Wasserstein convergence;
2. *Network design:* the Lipschitz constant $L_k(F)$ controls how embedding perturbations are amplified through the network;
3. *Bayes surrogate terms:* a residual contribution capturing the discrepancy between discrete labels and their Bayes predictor surrogate, including irreducible randomness.

## B Details of Experiments

### B.1 Details of Section 5.1

We conducted an experiment on MNIST to study how the Wasserstein distance between empirical embedding distributions depends on (i) the intrinsic dimension of the embeddings and (ii) the number of samples used to estimate the distributions.

**Model and training.** We trained simple fully connected autoencoders with a symmetric architecture. The encoder flattened each $28 \times 28$ image and mapped it to 256 hidden units with ReLU activation, followed by a linear layer to a $d$-dimensional bottleneck. The decoder mirrored this with a linear layer back to 256 units, ReLU, and a final linear layer to 784 units. Training used mean squared error loss, the Adam optimizer with learning rate $10^{-3}$, batch size 128, and 30 epochs. Global randomness was controlled by setting a fixed seed (2025) for both PyTorch and NumPy.

**Data and embeddings.** All data were drawn from the MNIST dataset. For the analysis of intrinsic dimension, we trained autoencoders with bottleneck sizes $d \in \{16, 32, 64, 128, 256, 512\}$. For the analysis of sample size, we trained a single autoencoder with bottleneck dimension 64 and repeatedly drew two independent subsets of size $n \in \{100, 200, \ldots, 1000\}$ to evaluate how the Wasserstein distance scales with $n$. In all cases, embeddings from the training split were used as the empirical distribution, and embeddings from the test split were used as the population distribution.

**Intrinsic dimension estimation.**    We estimated the intrinsic dimension using the maximum likelihood estimator of Levina and Bickel (Levina & Bickel, 2004), implemented in `skdim`.

**Wasserstein distance.**    We measured discrepancies between embedding sets using an entropically regularized optimal transport cost (Sinkhorn distance). Uniform weights were assigned to all points, the ground cost was the Euclidean distance, and the regularization parameter was $\varepsilon = 10^{-2}$. Iterations terminated either after 200 steps or once the update magnitude fell below $10^{-6}$. The resulting cost was computed as the expectation of the ground cost under the transport plan.

### B.2    DETAILS OF SECTION 5.2

We conducted experiments on CIFAR-10 and CIFAR-100 to analyze how the final-layer embeddings relate to class-wise generalization gaps modified across ResNet architectures.

**Model and training.**    We considered five ResNet architectures: ResNet-18, 34, 50, 101, and 152. Each model was initialized with ImageNet-pretrained weights from `torchvision.models` and evaluated on CIFAR datasets. The architecture of these ResNet models was modified by adjusting the final linear output layer. Specifically, the output of the model's convolutional layers is initially projected to a 128-dimensional space via a linear layer. This is then followed by a Sigmoid activation function, and finally, another projection layer yields the ultimate output. These nets are finetuning on Cifar-10 and Cifar-100 used the Adam optimizer with weight decay 0.001, base learning rate $10^{-4}$, and a cosine annealing schedule over 50 epochs. Batch size was 256, with random horizontal flip for augmentation. Multi-GPU training was enabled via `accelerate`. Models were saved after training and evaluated on the full test set.

**Embedding extraction.**    For each trained model, we extracted the *last layer embeddings* for all samples in both training and test splits. Embeddings were stored separately for each class to allow class-wise analysis. For the CIFAR-10 dataset, each class of embeddings in both the training and test sets comprises 500 samples. In the case of the CIFAR-100 dataset, each class of embeddings in both the training and test sets consists of 100 samples.

**Intrinsic dimension estimation.**    We estimated the intrinsic dimension of these embeddings using the maximum likelihood estimator of Levina and Bickel (Levina & Bickel, 2004), as implemented in `skdim`. Estimates were computed independently for each class and averaged across samples, yielding 10 estimates per model on CIFAR-10 and 100 per model on CIFAR-100.

**Wasserstein distance.**    For each class, we computed the empirical 1-Wasserstein distance between training and test embeddings. This used entropic-regularized optimal transport (Sinkhorn distance) with Euclidean ground cost, uniform weights, and regularization parameter $\epsilon = 10^{-2}$. These distances quantify how far apart the validation and test embedding distributions are.

**Generalization gap.**    For each class and model, validation and test losses were recorded to compute the class-wise generalization gap.

### B.3    DETAILS OF SECTION 5.3

We designed a experiment on MNIST to analyze how the width of a hidden layer influences intrinsic dimension of intermediate embeddings, Lipschitz properties of the network and generalization performance. The experiment uses a six-layer multilayer perceptron (MLP) with configurable hidden-layer widths and records both statistical and geometric properties of representations throughout training.

**Model and training.**    The model is a fully connected network with architecture

$$784 \rightarrow h_1 \rightarrow h_2 \rightarrow h_3 \rightarrow h_4 \rightarrow h_5 \rightarrow 10,$$

where each hidden layer is followed by a ReLU activation. The default hidden width is 100 units for all layers. To study the effect of representation bottlenecks, we varied the width of the third hidden

layer ($h_3$) over the list $\{100, 90, 80, 70, 60, 50, 40, 30, 20, 10\}$, while keeping all other layers fixed at 100. Training was performed with cross-entropy loss, the Adam optimizer (learning rate $10^{-3}$, weight decay 0), batch size 128, and for 10 epochs. We used both training and test splits of MNIST, with additional evaluation on a fixed random subset of 2048 training samples. All randomness was controlled by fixed seeds and deterministic settings in PyTorch to ensure reproducibility.

**Activation collection and intrinsic dimension.** To measure representation complexity across layers we registered forward hooks after each ReLU activation. During evaluation, the hooks collected activations for all inputs in the 2048-sample subset. For each layer's activation matrix $X$, we applied the maximum likelihood estimator (as implemented in `skdim`).

**Lipschitz estimation.** To characterize stability of the mapping from each hidden layer to the output, we computed the product of spectral norms of all subsequent linear layers. For a given suffix starting at layer $i$, the Lipschitz constant was approximated by

$$L_{i \to \text{end}} = \prod_{j=i+1}^{L} \sigma_{\max}(W_j),$$

where $W_j$ denotes the weight matrix of linear layer $j$ and $\sigma_{\max}$ is its top singular value. Singular values were computed using `torch.linalg.svdvals` in double precision. These suffix-wise Lipschitz estimates were recorded at initialization and after each epoch.

## C  DIMENSIONALITY ESTIMATION AND HYPERPARAMETER ANALYSIS

In this appendix, we investigate the effects of hyperparameter choices and the specific algorithm used on the estimation of embedding dimensionality. All experiments are conducted using subsets of the CIFAR datasets: 500 samples per class for CIFAR-10 and 100 samples per class for CIFAR-100.

### C.1  HYPERPARAMETER ANALYSIS

We first examine how the choice of the hyperparameter $K$ affects dimensionality estimates. Here, $K$ corresponds to the number of nearest neighbors used in the estimation procedure: larger $K$ values capture dimensionality over a broader range of the data, whereas smaller $K$ values reflect more local structure.

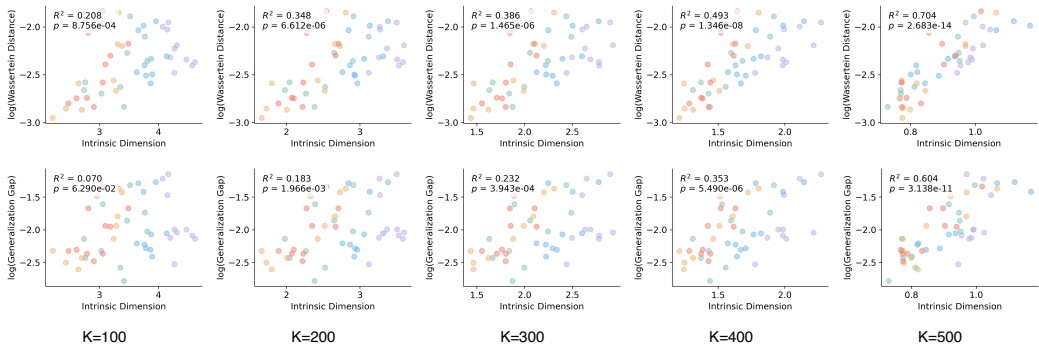

Figure 5: **Effect of hyperparameter $K$ on dimensionality estimation for CIFAR-10 embeddings.** Larger $K$ values capture broader data structure and lead to higher correlation with generalization error.

For CIFAR-10, we test $K = 100, 200, 300, 400, 500$, and for CIFAR-100, we test $K = 20, 40, 60, 80, 100$. Figures 5 and 6 show the results. We observe that as $K$ increases, the estimated dimensionality better correlates with the generalization error. This indicates that the global dimensionality of a class is more predictive of generalization performance than local dimensionality.

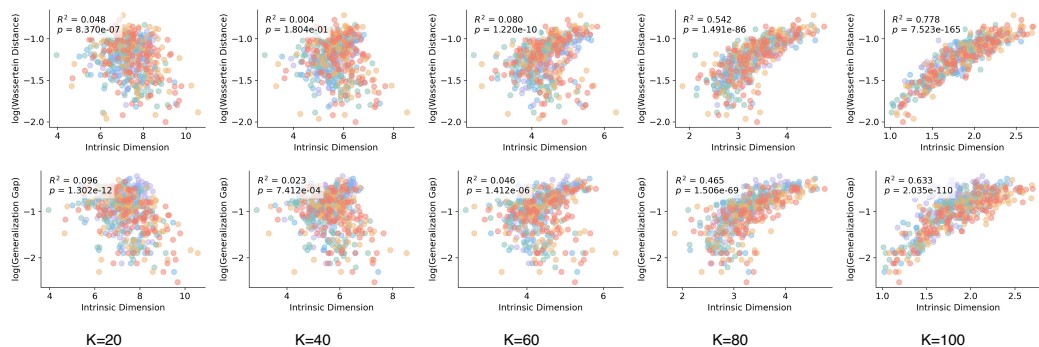

Figure 6: **Effect of hyperparameter $K$ on dimensionality estimation for CIFAR-100 embeddings.** Increasing $K$ improves the alignment between estimated dimensionality and generalization error, indicating that global structure is more informative.

## C.2 ALGORITHM COMPARISON

Next, we compare different dimensionality estimation algorithms (TLE (Amsaleg et al., 2019) and MOM (Amsaleg et al., 2018)) while keeping the hyperparameter fixed ($K = 400$ for CIFAR-10, $K = 80$ for CIFAR-100).

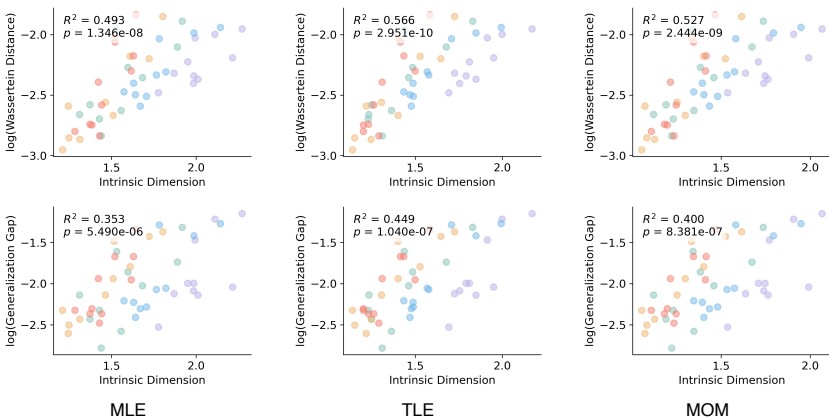

Figure 7: **Comparison of dimensionality estimation algorithms on CIFAR-10 embeddings.** Despite using different algorithms, estimated dimensionalities consistently correlate with generalization error, demonstrating robustness to method choice.

Figures 7 and 8 present the results. Across both datasets, all algorithms yield estimated dimensionalities that remain significantly correlated with generalization error, suggesting that the observed relationship is robust to the choice of estimation method.

## C.3 ANALYSIS USING ALL SAMPLES

Section 5.2 of the main paper analyzes each class independently. Here we complement that analysis by examining all validation samples pooled together, in order to assess whether the cross-model trends observed at the class level also manifest at the level of the entire dataset.

We first compute the intrinsic dimension using all samples jointly. As shown in Figure 9, the dimension estimated from the full dataset remains strongly correlated with the generalization error. This trend is consistent across both CIFAR-10 and CIFAR-100, demonstrating that the representation geometry at the dataset level preserves the same predictive relationship observed at the class level.

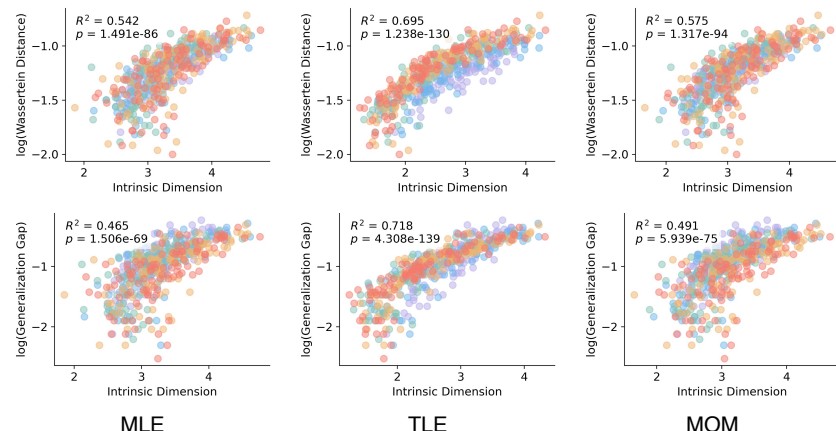

Figure 8: **Comparison of dimensionality estimation algorithms on CIFAR-100 embeddings.** Dimensionality estimates remain significantly associated with generalization error across different algorithms.

We also examined an alternative procedure in which the per-class intrinsic dimensions are first computed and then averaged. This averaging substantially weakens the correlation, especially for CIFAR-100. The primary reason is that the intrinsic dimensions of different classes vary considerably, so simple averaging fails to capture the true geometric complexity of the overall data distribution. These results suggest that the more principled approach is to estimate intrinsic dimension directly from all samples, rather than aggregating per-class estimates.

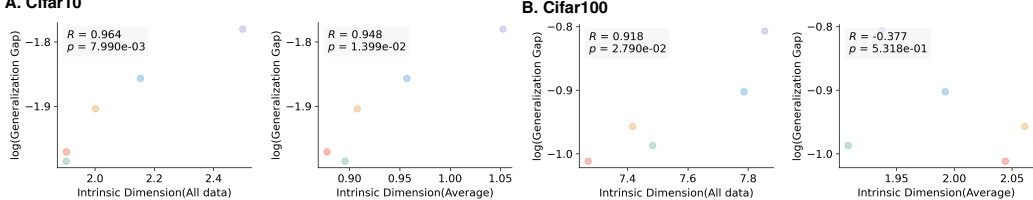

Figure 9: **Intrinsic dimension computed across all samples predicts generalization.** (A) CIFAR-10 and (B) CIFAR-100 show that the intrinsic dimension estimated from the pooled validation set exhibits a strong correlation with generalization error. In contrast, the mean of per-class dimensions leads to weaker correlations, particularly on CIFAR-100, where class-wise variability is large.

## C.4 SINGLE-MODEL ANALYSIS

The results in Section 5.2 pool together all architectures and all classes. To confirm that the observed relationships do not arise solely from cross-model variability, we additionally analyze each architecture in isolation.

Figures 10 and 11 show the results for CIFAR-10 and CIFAR-100, respectively. For every architecture, the intrinsic dimension and the Wasserstein distance computed at the final layer both remain strongly correlated with the generalization error. These results indicate that the geometricstatistical relationship predicted by our theory holds not only across different architectures, but also within each individual model.

## C.5 ANALYSIS OF LARGE-SCALE PRETRAINED MODELS

We further extend our analysis to a set of large-scale ConvNeXt models with diverse pretraining regimes, including ImageNet-1K, ImageNet-22K, and LAION-based CLIP-style pretraining. The specific models evaluated are:

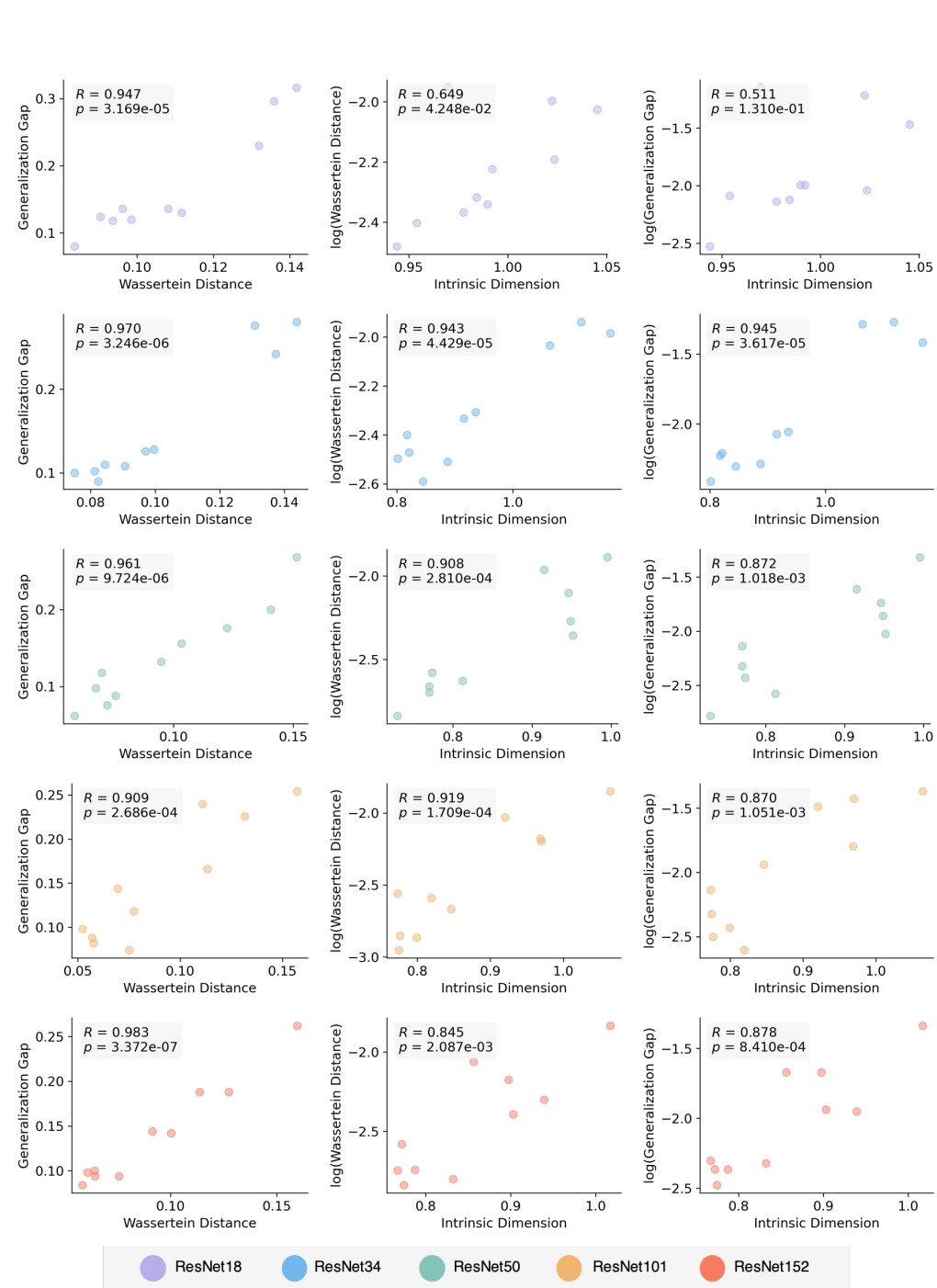

Figure 10: **Within-model relationships on CIFAR-10.** For each architecture analyzed independently, both intrinsic dimension and Wasserstein distance computed from the final-layer embeddings correlate strongly with generalization error, confirming that the relationship holds at the single-model level.

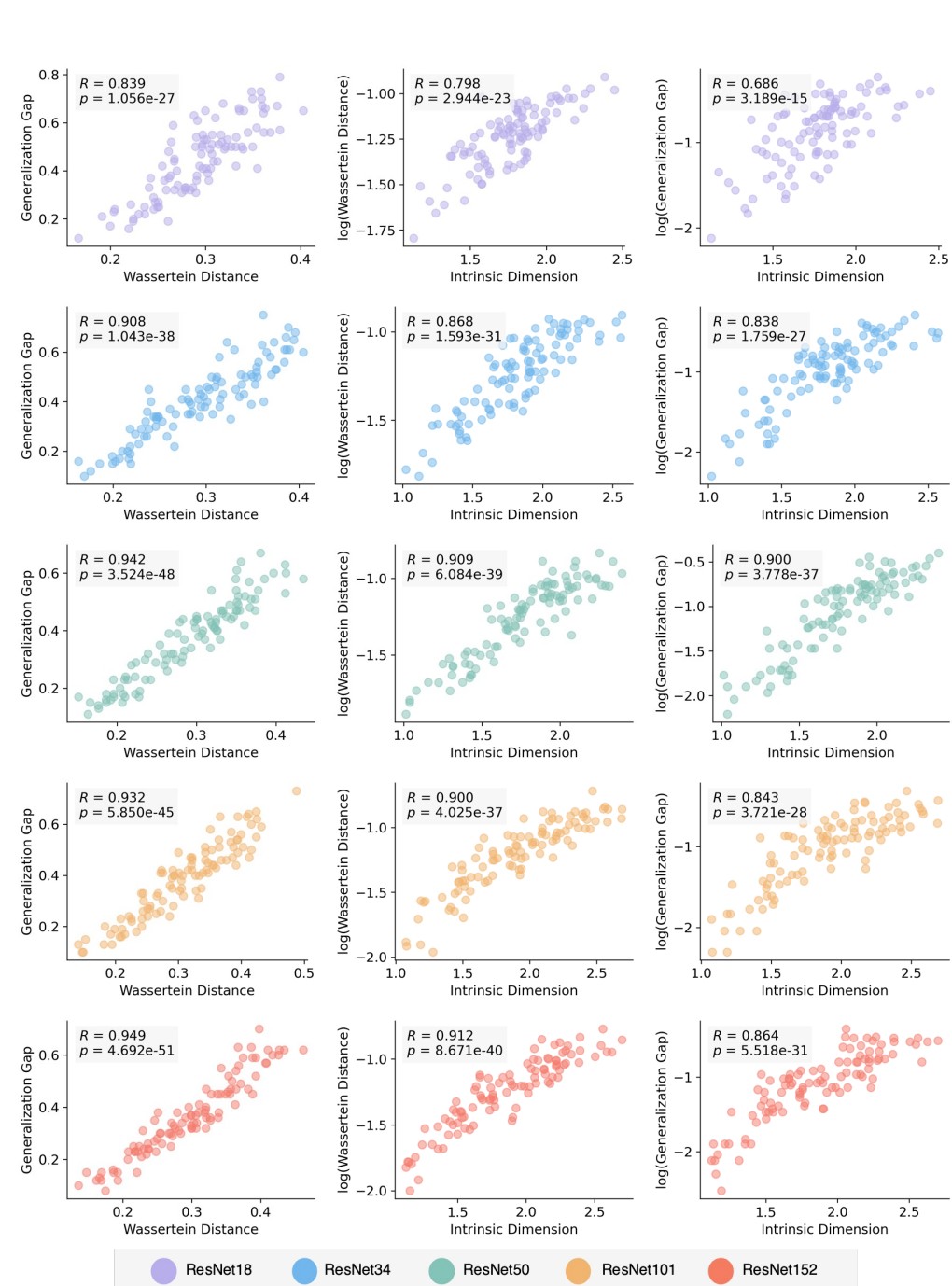

Figure 11: **Within-model relationships on CIFAR-100.** The correlation patterns between intrinsic dimension, Wasserstein distance, and generalization error remain consistent when evaluating each architecture individually. This demonstrates that the geometric predictors identified by our analysis apply robustly across datasets and model scales.

```
     convnext_base.clip_laion2b_augreg_ft_in1k,
    convnext_base.clip_laiona_augreg_ft_in1k_384,
    convnext_base.fb_in22k_ft_in1k, convnext_base.fb_in22k_ft_in1k_384,
    convnext_large.fb_in22k_ft_in1k, convnext_large.fb_in22k_ft_in1k_384,
    convnext_large_mlp.clip_laion2b_augreg_ft_in1k,
    convnext_large_mlp.clip_laion2b_augreg_ft_in1k_384,
    convnext_nano.in12k_ft_in1k, convnext_small.fb_in22k_ft_in1k,
    convnext_small.fb_in22k_ft_in1k_384, convnext_small.in12k_ft_in1k,
    convnext_small.in12k_ft_in1k_384, convnext_tiny.fb_in22k_ft_in1k,
    convnext_tiny.fb_in22k_ft_in1k_384, convnext_tiny.in12k_ft_in1k,
    convnext_tiny.in12k_ft_in1k_384
```

Despite the substantial scale of these models and the heterogeneity of their pretraining datasets, the relationship between intrinsic dimension, Wasserstein distance, and generalization performance remains consistent. Figure 12 summarizes the results. These findings demonstrate that the predictive power of embedding geometry extends to modern large-scale models and high-capacity pretraining regimes.

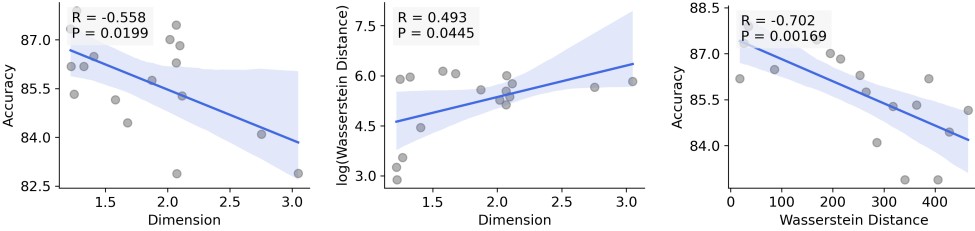

Figure 12: **Large-scale pretrained ConvNeXt models exhibit strong correlations among intrinsic dimension, Wasserstein distance and generalization performance.** Across a wide range of ConvNeXt variants, both the intrinsic dimension and the Wasserstein distance remain strongly correlated with generalization performance on ImageNet classification. These results indicate that this geometric–generalization relationship persists in large models and large-scale datasets.

### C.6 LAYER-WISE CORRELATIONS AMONG DIMENSION, WASSERSTEIN DISTANCE AND GENERALIZATION PERFORMANCE

We analyzed embeddings from ResNet-152 at layers 4, 18, 30, 43, 55, 67, 79, 91, 103, 115, 127, 139, and 152, and computed the correlation between embedding dimensionality, Wasserstein distances on the validation and test sets, and generalization error.

Correlations are relatively weak in early layers but increase in deeper layers, with a pronounced rise after layer 140. This suggests that deeper embeddings more faithfully capture features relevant to generalization.

### C.7 DYNAMICS OF EMBEDDING DIMENSIONALITY DURING TRAINING

We trained a simple convolutional network on CIFAR-10 and tracked the dimensionality of the final-layer embeddings throughout training.

Dimensionality initially decreases and then rises, rather than continuously declining. This behavior is expected, if dimensionality were to decrease monotonically, the observed correlation between dimensionality and generalization error would fail to account for overfitting phenomenon.

Moreover, embedding dimensionality is nearly identical across training, validation and test sets, indicating that either training or validation embeddings can reliably reflect the overall data distributions representational structure.

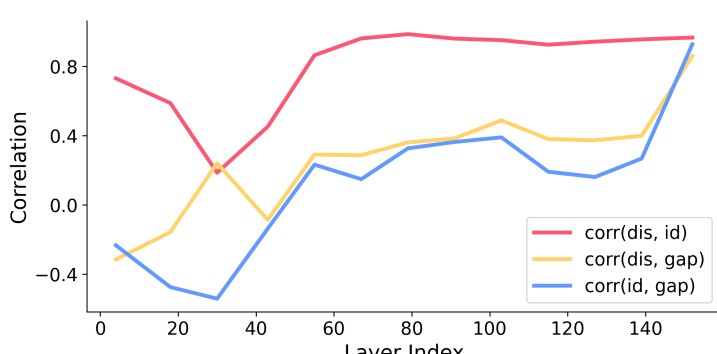

Figure 13: **Layer-wise correlations between embedding dimensionality, Wasserstein distance, and generalization error in ResNet-152.** Deeper layers exhibit stronger correlations, indicating the increasing alignment between representation properties and generalization.

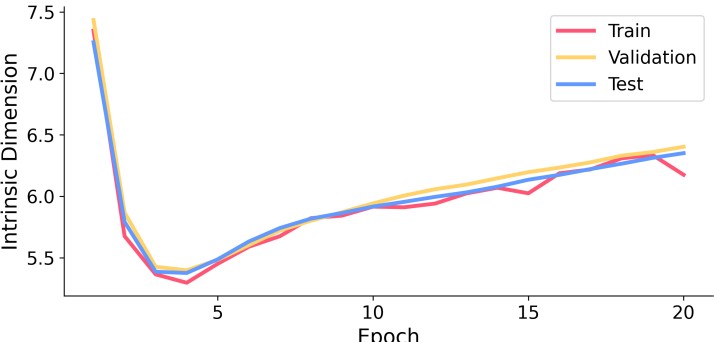

Figure 14: **Training dynamics of embedding dimensionality.** Dimensionality decreases in early training and rises later, reflecting its relationship with overfitting and generalization.

## D   LLM Usage Statement

In accordance with the ICLR 2026 policy on responsible usage of Large Language Models (LLMs), we disclose that LLMs were employed to aid in the preparation of this manuscript. Specifically, LLMs were used to polish writing, improve clarity and refine grammar. All ideas, analyses and conclusions presented in this work are solely those of the authors.

