# OpenReview forum: "Generalization Error Bound via Embedding Dimension and Network Lipschitz Constant"
_ICLR.cc/2026/Conference — ICLR 2026 Conference Withdrawn Submission_

### Official Review · Reviewer_sKHb · 2025-10-26

**Soundness:** 3
**Presentation:** 2
**Contribution:** 3
**Rating:** 6
**Confidence:** 3

**Summary:**

This paper advances a representation-centric view of generalization, deriving a bound that depends on (i) the intrinsic dimension of learned embeddings and (ii) the network’s Lipschitz sensitivity from embeddings to outputs. Lower-dimensional embeddings yield faster empirical-to-population convergence in Wasserstein distance (scaling $\approx n^{-1/d}$), while smaller Lipschitz constants limit error amplification. At the final layer, architectural sensitivity vanishes, so the bound is driven primarily by embedding dimension—explaining why final-layer dimensionality often tracks generalization. Experiments support the theory: an MNIST autoencoder matches the predicted scaling; across ResNet-18/34/50/101/152 on CIFAR-10/100, final-layer dimension and train–test Wasserstein distance correlate with the generalization gap. A width-reduction intervention lowers embedding dimension but raises Lipschitz sensitivity, yielding no consistent gain—confirming the joint role of both factors. Overall, the work links representation geometry with architecture, explains the prominence of final-layer dimension, and offers actionable diagnostics (e.g., monitoring embedding dimension) for practice.

**Strengths:**

- Novel framework linking representation geometry and sensitivity: The paper derives a representation-centric generalization bound that depends on (i) the intrinsic dimension of embeddings and (ii) the network’s Lipschitz sensitivity. This bridges model architecture and data distribution, offering a more interpretable alternative to parameter-based capacity measures. Building on sharp Wasserstein convergence rates, it provides a concrete theoretical link between lower-dimensional learned features and improved generalization, aligning with recent empirical observations.

 - Insight into final-layer representations and generalization: A key insight from the theory is that at the final embedding layer, architectural sensitivity drops out of the bound, leaving intrinsic dimension as the dominant driver of generalization. This explains why last-layer dimensionality reliably predicts performance, and the authors corroborate it empirically (ResNets on CIFAR): lower final-layer dimension → smaller generalization gap. The result offers an intuitive geometric diagnostic that often outperforms parameter-based measures.

 - Comprehensive empirical validation.
The experiments substantiate the theory on three fronts: (1) Scaling law: an MNIST autoencoder shows Wasserstein distance grows with embedding dimension and shrinks with sample size, matching the $\approx n^{-1/d} trend; (2) Cross-architecture correlation: on CIFAR-10/100 (class-wise), final-layer intrinsic dimension and train–test embedding divergence strongly correlate with the generalization gap across ResNet-18/34/50/101/152; (3) Causal probe: narrowing an intermediate MLP layer lowers embedding dimension but increases Lipschitz sensitivity (via spectral-norm proxy), yielding no consistent generalization gains, confirming both factors jointly matter. Results are consistent across datasets and architectures, indicating the proposed representation-based metrics are robust and broadly relevant.

**Weaknesses:**

- Strong assumptions: The derivation of the bounds involves certain non-trivial assumptions that may limit the scope of the theory. In particular, the analysis assumes that the Bayes-optimal predictor is Lipschitz continuous with respect to the learned embedding space. This assumption is necessary to ensure a well-behaved relationship between changes in the embedding and changes in the target, essentially bounding the “irreducible” part of the problem. While this makes the mathematics tractable, it might be unrealistic in some practical tasks that the true data generating process might not be Lipschitz, especially in high-dimensional raw input space or if classes have very complex decision boundaries. Thus, the necessity of this condition somewhat narrows the situations where the bound is theoretically guaranteed. It would be interesting to know how sensitive the conclusions are to this assumption or if it can be relaxed.

- Loose bounds and unclear tightness: A potential limitation of the proposed generalization bound is that it comes with unspecified or potentially loose constant factors. The theory introduces constants (e.g. $C_k, D_k, \varepsilon$ in the bound) and relies on order-of-magnitude terms (like $n^{-1/(d+\varepsilon)}$), which means the numerical value of the bound might be quite loose or vacuous for practical network sizes. The authors themselves acknowledge that “our bound contains constants that may be loose”. As a result, while the bound is qualitatively insightful (e.g. predicting which direction generalization will move when intrinsic dimension changes), it might not be quantitatively tight enough to certify performance or compare different models in absolute terms. Indeed, the evidence provided is mostly correlational rather than showing the bound as an exact predictive formula.

 - Practical measurement of complexity terms: From a practical perspective, one challenge is that the two key quantities in the bound, the intrinsic dimension $d_k$ of embeddings and the network’s Lipschitz constant $L_k$, are not trivial to measure or control in real-world settings. Estimating intrinsic dimension of embeddings was done via a k-NN based Maximum Likelihood Estimator in the experiments, which introduces some estimation error, though the authors did cross-validate this with multiple methods. More problematic is the Lipschitz constant of a deep network, which is difficult to compute exactly. The paper turns to an upper bound as an approximation in experiments, but this can be a very loose over-estimate of the true Lipschitz constant, especially for networks with many layers or nonlinear activations. This means the absolute value of the sensitivity term is hard to pin down, limiting the ability to use the bound quantitatively. The authors explicitly note that “estimating the sensitivity term $L_k$ remains challenging in practice, and developing reliable estimators is necessary for broader applicability”.

 - Relationship between factors: The theory treats the intrinsic dimension and the Lipschitz sensitivity as if they were independent factors, but in practice they can be intertwined. For instance, architectural changes can simultaneously affect both, for example, a narrower layer lowered the dimension but raised the Lipschitz constant. The authors acknowledge this as well, stating that in the current analysis they treat dimension and sensitivity independently, whereas “in reality, these quantities may be correlated,” and understanding their interplay is an important avenue for future work. This interplay makes it non-trivial to directly apply the bound for model improvements: one cannot simply minimize one factor (e.g. enforce a small embedding dimension) without considering potential adverse effects on the other (increased sensitivity). The paper stops short of providing guidance on how to jointly manage these factors.

 - Limitation in empirical validations: It’s worth noting that the empirical evaluation, though extensive within the scope of the paper, is concentrated on vision classification tasks with relatively moderate-sized datasets (MNIST, CIFAR) and standard architectures. It remains an open question how well these representation-based generalization insights carry over to much larger-scale settings (e.g. ImageNet-scale vision models or NLP tasks) or more complex scenarios, which is a relatively important empirical exploration.

**Questions:**

1. The theoretical analysis assumes the Bayes-optimal predictor is Lipschitz continuous with respect to the embedding space. How realistic is this assumption in practice, and what intuition can the authors provide about when it would hold? For example, are there arguments or evidence that, at least for learned final-layer features, the class label function is approximately Lipschitz? If this assumption were violated, how would it affect the applicability of the bound? It would be helpful if the authors could elaborate on the conditions under which this Lipschitz assumption is justified or provide potential idea on relaxing it.

2. The proposed bound involves unspecified (potentially loose) constants (e.g., $C_{k},D_{k},M_{F},M_{F^{*}},L_{k}(F),L_{k}(F^{∗})$) and an asymptotic rate $n^{−1/(d+\varepsilon)}$, raising concerns about quantitative usefulness. For at least one setting, could you instantiate the bound numerically to show whether it is non-vacuous? How sensitive are conclusions to $\varepsilon$, and can it be data-driven (via local covering behavior) rather than a free slack? Are there identifiable regimes (ranges of
$d_{k},L_{k},n$) where the bound is guaranteed to be below observed gaps, or a calibration strategy that makes the bound quantitatively predictive across architectures rather than merely correlational?

3. Could the authors clarify how they define the intrinsic dimension of learned embeddings in formal terms, and discuss the reliability of its estimation? The paper uses an MLE method to estimate $d_k$. How sensitive are the results to the choice of dimension estimation technique or to the sample size used for estimation? For high-dimensional embeddings or limited data, do the authors observe any issues with accurately measuring intrinsic dimension? Similar questions for the Lipschitz constant.

4. The width intervention experiment nicely illustrates that simply forcing a smaller embedding dimension can be counterproductive due to the rise in Lipschitz constant. This suggests a trade-off scenario. Could the authors provide more insight into this interplay? For example, do certain architectural choices or layer types inherently produce lower-dimensional embeddings without as large an increase in sensitivity (perhaps residual connections, or using autoencoder-like bottlenecks with regularization)? The paper mentions that treating dimension and sensitivity as independent is a simplification. How might one theoretically approach the scenario where $d_k$ and $L_k$ are correlated? Is there a version of the bound or an analysis that accounts for their joint distribution rather than a simple product of separate terms? Any thoughts on promising directions to address this coupling would be appreciated, as it seems key to turning these insights into concrete design principles for neural networks.

5. Have the authors considered evaluating their representation-based generalization metrics on other domains or larger-scale tasks? How do the authors envision applying their analysis to the next scale of problems, and do they anticipate any new phenomena or difficulties might arise there? For instance, would measures like embedding intrinsic dimension also correlate with generalization in NLP models or on ImageNet-scale vision models? The current results use CIFAR and relatively standard networks; exploring whether the same trends hold for, for example, transformer models in NLP or very deep networks on large datasets would strengthen confidence that these insights are universal. Any preliminary observations or thoughts on potential challenges (e.g. intrinsic dimension estimation in very large networks, or the effect of tasks with many classes) would be interesting.

---

> ### Author Response · Authors · 2025-11-27
>
> > 1. Strong assumptions: The derivation of the bounds involves certain non-trivial assumptions that may limit the scope of the theory.
>
> We adopt this assumption because, without any smoothness assumption on the Bayes predictor, the mapping from embeddings to labels could vary arbitrarily in any local neighborhood (e.g., very close embeddings could correspond to completely different labels). In such cases, using only the statistical and geometric properties of embeddings (e.g., intrinsic dimension) would not suffice to control the propagation of risk from the empirical to the population distribution, making it impossible to provide any meaningful generalization bound. In other words, the Lipschitz constant of the Bayes predictor acts as a measure of the “task difficulty” or label noise in our bound.
>
> In practice, this assumption is approximately satisfied in many scenarios: for embeddings in the final layer or close to the output, features often encode high-level semantic information smoothly, making the class-conditional expectation relatively smooth. Additionally, for networks with ReLU activations and linear heads, the Lipschitz constant can be upper-bounded by the product of spectral norms of the parameter matrices.
>
> > 2. Loose bounds and unclear tightness: A potential limitation of the proposed generalization bound is that it comes with unspecified or potentially loose constant factors.
>
> We agree that the theoretical bound contains constants that are not fully explicit, and thus may be loose quantitatively. However, the primary goal of this work is to reveal how embedding dimension affects generalization error, rather than to construct a numerically tight bound. Across extensive experiments, embedding dimension consistently emerges as the dominant factor influencing generalization error: the correlation between embedding dimension and generalization gap remains significant regardless of model scale. This indicates that the leading term in the bound, the dimension-dependent Wasserstein convergence rate, is indeed the main driver, rather than the potentially large constants.
>
> > 3. Practical measurement of complexity terms: From a practical perspective, one challenge is that the two key quantities in the bound, the intrinsic dimension of embeddings and the network’s Lipschitz constant , are not trivial to measure or control in real-world settings.
>
> For intrinsic dimension, we employed multiple estimation methods (MLE, TLE, MOM) both in the main text and appendix C.2, and observed high consistency in trends and correlation.
>
> We acknowledge that estimating the Lipschitz constant of a deep network remains challenging. However, our theoretical results show that when analyzing the embeddings at the last layer, the effect of the network’s Lipschitz constant can be eliminated. Therefore, even if the Lipschitz constant cannot be precisely estimated, it does not compromise the practical applicability of our theoretical results.
>
>
> > 4. Limitation in empirical validations:It remains an open question how well these representation-based generalization insights carry over to much larger-scale settings (e.g. ImageNet-scale vision models or NLP tasks) or more complex scenarios, which is a relatively important empirical exploration.
>
> We have supplemented **Appendix C.5** with experiments on large pre-trained models and ImageNet-scale datasets. Despite the increased model and data scale, we observe the same trend: embedding dimension remains significantly correlated with generalization error.
>
> > 5. The theoretical analysis assumes the Bayes-optimal predictor is Lipschitz continuous with respect to the embedding space. How realistic is this assumption in practice, and what intuition can the authors provide about when it would hold?
>
> In Corollary 1, we note that for the final output layer, there exists an identity mapping from the network output to the labels, which has a Lipschitz constant of 1. Hence, for the output layer, this assumption holds.
>
> Furthermore, networks with ReLU activations are themselves Lipschitz continuous, and their Lipschitz constant can be controlled by the product of the spectral norms of their parameter matrices. Therefore, minimizing the spectral norms can help reduce the network Lipschitz constant in practice.
>
>
> > 6. The proposed bound involves unspecified (potentially loose) constants and an asymptotic rate , raising concerns about quantitative usefulness.
>
> We agree that the current bound primarily provides a structural explanation rather than precise quantitative estimates. Its main value is to reveal how embedding dimension and the network’s Lipschitz constant jointly constrain generalization. Additionally, if the constants were extremely large, changes in embedding dimension would not significantly correlate with changes in generalization error. Our experiments with both small and large models support that the dimension-related term indeed dominates.

---

> > ### Author Response · Authors · 2025-11-27
> >
> > > 7. Could the authors clarify how they define the intrinsic dimension of learned embeddings in formal terms, and discuss the reliability of its estimation? The paper uses an MLE method to estimate d_k. How sensitive are the results to the choice of dimension estimation technique or to the sample size used for estimation? For high-dimensional embeddings or limited data, do the authors observe any issues with accurately measuring intrinsic dimension? Similar questions for the Lipschitz constant.
> >
> > We provide a formal definition of intrinsic dimension in Section 3.4, using the upper Wasserstein dimension to characterize the effective geometric complexity of the main probability mass of the embedding distribution.
> >
> > Estimating with 500 samples is sufficient to obtain a significant correlation with generalization error across most datasets. Appendix C.2 compares different estimators (MLE, TLE, MOM), showing consistent results.
> >
> > In Appendix C.5, we further analyze embeddings from large-scale pre-trained models and downstream task generalization errors. These embeddings were reduced to 300 dimensions via PCA. Despite higher ambient dimension and data complexity, we still observe significant correlations, suggesting that with sufficient samples, the ambient dimension has little impact.
> >
> > Estimating the Lipschitz constant remains a challenge. While for ReLU networks we can compute an upper bound via the product of spectral norms, this may still be loose. More accurate estimation methods will be needed in the future.
> >
> > > 8. Could the authors provide more insight into this interplay? For example, do certain architectural choices or layer types inherently produce lower-dimensional embeddings without as large an increase in sensitivity (perhaps residual connections, or using autoencoder-like bottlenecks with regularization)?
> >
> > The interplay between network architecture, embedding dimension, and Lipschitz constant is indeed a complex topic. However, as mentioned in Corollary 1, when focusing on the final-layer embeddings, the effect of the network’s Lipschitz constant can be eliminated. Therefore, one can optimize the network’s output to reduce embedding dimension and improve overall performance without being limited by the Lipschitz term.
> >
> > > 9. Have the authors considered evaluating their representation-based generalization metrics on other domains or larger-scale tasks?
> >
> > In Appendix C.5, we supplement analysis of pre-trained models and ImageNet-scale datasets. The results demonstrate that, even in large models and complex datasets, embedding dimension, Wasserstein distance, and generalization error remain significantly correlated. This suggests that our representation-based generalization insights extend beyond CIFAR and standard networks, and may hold in larger-scale vision and potentially NLP tasks.

---

### Official Review · Reviewer_dY4n · 2025-10-27

**Soundness:** 2
**Presentation:** 3
**Contribution:** 2
**Rating:** 4
**Confidence:** 4

**Summary:**

This paper proposes new generalization bounds for neural networks. The proposed bounds apply to every hypothesis in the parameter space and are based on the intrinsic dimension of an intermediate layer (which can be the last) and on the Lipschitz constant of the subsequent layers. Given a parameter vector and a dataset of iid data points, the main technical idea is analyse the Wasserstein distance between the pushforward of both the data distribution and the empirical distribution of the dataset by the first $k$ layers of the network. Then, Kantorovitch-Rubinstein duality is applied by exploiting the regularity of the subsequent layers. This proof technique is based on state of the art results for the estimation of the Wasserstein distance between a probability distribution and its empirical counterpart. This introduces a notion of intrinsic dimension that the authors relate to the neural network architecture to obtain their final bounds, based on embedding dimensions. The theory is supported by several experiments.

**Strengths:**

- The paper finely exploits the regularity of the neural network architecture, which is a factor that is important to take into account in generalization bounds.
- The proposed bounds are computable. In particular, the new notion of intrinsic dimension is interesting and the experiments are conducted in various settings.
- The application of the proposed theory at the last layer is particularly interesting, as the regularity conditions are easily obtained. This supports the empirical correlation between the dimension of the last layer and the generalization error.

**Weaknesses:**

*Main weaknesses:*
 - The main bound (Thm 4.1) holds in high probability for a fixed predictor (in particular, F seems to be independent of the data). As it is assumed that the loss is bounded, in this setting a simple application of Hoeffding inequality gives a bound of order $\sqrt{\log(1 / \delta) / n}$ on $\mathrm{gen}(F)$, which seems to have a better rate than you bound unless the intrinsic dimension is strictly smaller than 2. I believe this requires some additional comments.
 - In practice, the predictor is not fixed, but it is trained based the training dataset. If such a data-dependent predictor is used, then it is not true anymore that the push forward of the empirical distribution at layer $k$ corresponds to the empirical distribution associated with the embedding distribution (because the points are not independent anymore). Therefore, the proposed proof technique would not apply. It seems to be an important issue as obtaining a bound for fixed predictor can be done with classical concentration inequalities, a new high probability generalization bound should be able to apply to data-dependent predictors.
 - If think the paper "Instance-Dependent Generalization Bounds via Optimal Transport" (Hou et al., 2022) should be added to the discussion of related works, as it also uses Wasserstein distances and intrinsic dimensions.


*Other issues:*
 - In definition 7, the notion of effective dimension does not seem to have been defined before.
 - the notation $\mathrm{gen}$ is also used before it is defined.
 - in Theorem 3.1, $d_p^\star$ has not been defined, which makes this theorem hard to read.

**Questions:**

- Can you explain what is the exact role of the neighbourhood $U_k$ in Assumption 3?
- In what cases can the intrinsic dimension of the embedding distribution be much smaller than the number of neurons in the layer.
- Can you produce the same plot as Fig. 3, but using the number of neurons in the intermediate layer instead of the intrinsic dimension estimator?
- Can you comment on why the reported intrinsic dimensions are quite small on Fig. 3?
- In addition to Figure 3, could you compute correlation coefficients (eg, Kendall) to get a more precise understanding of the observed correlation?

---

> ### Author Response · Authors · 2025-11-27
>
> > 1. The main bound (Thm 4.1) holds in high probability for a fixed predictor (in particular, F seems to be independent of the data). As it is assumed that the loss is bounded, in this setting a simple application of Hoeffding inequality gives a bound of order $\sqrt{(1/δ)/n}$ on gen(F), which seems to have a better rate than you bound unless the intrinsic dimension is strictly smaller than 2.
>
> Assuming Hoeffding’s inequality, the bound would be:
>
> $ |R(F) - \hat R_n (F)| \le B_l \sqrt{\frac{1}{n} log {2}{\delta} } $
>
> This bound depends on the upper bound of the loss $B_l$ , which can be very large, making it quite loose in practice. In other words, the faster rate given by Hoeffding’s inequality relies on a very coarse worst-case constraint. In contrast, our bound does not depend on an upper bound of the loss and aims to explain why different models on the same dataset can exhibit different generalization behavior depending on their embedding dimension. Therefore, the two types of bounds serve different purposes and are valuable in their respective contexts.
>
> > 2. In practice, the predictor is not fixed, but it is trained based the training dataset. If such a data-dependent predictor is used, then it is not true anymore that the push forward of the empirical distribution at layer k corresponds to the empirical distribution associated with the embedding distribution.
>
> We thank the reviewer for this important remark. In our theoretical analysis, the bound only requires analyzing the properties of empirical distribution (dimension and Wasserstein distance), without direct analysis of the training set. Therefore, the theory itself does not need modification. However, in experiments, there is indeed a dependency between the training set and the predictor, which would violate the independence assumption. To address this, in all theory-related estimations (including Wasserstein convergence and intrinsic dimension), we use embeddings from a validation set. The validation set is independent of the predictor, ensuring the empirical distribution is strictly iid, and thus the derivations fully hold.
>
> > 3. If think the paper "Instance-Dependent Generalization Bounds via Optimal Transport" (Hou et al., 2022) should be added to the discussion of related works, as it also uses Wasserstein distances and intrinsic dimensions.
>
> We thank the reviewer for pointing this out. Our analysis is indeed conceptually related, as both are inspired by Weed & Bach (2019) and exploit the effect of dimension on Wasserstein convergence to study generalization. However, the key difference is that Hou et al. focus on the raw data and its dimension, whereas we focus on model embeddings. As a result, their analysis explains how dataset complexity affects generalization, while our analysis explains how the complexity of representation learned by the model influences generalization. This allows us to understand why models with stronger compression capabilities generalize better and why the last-layer embedding dimension can predict generalization performance [2].
>
> From another perspective, our work complements theirs, providing a full picture from input data to hidden and output embeddings, showing how dimension at each stage affects the Wasserstein distance between empirical and population distributions, and thereby generalization performance.
>
> > 4. In definition 7, the notion of effective dimension does not seem to have been defined before.
>
> We provide a formal definition in Section 3.4 in the revised paper. Effective dimension is defined using the Upper Wasserstein Dimension, which quantifies the complexity of the main part of the data distribution. Conceptually, this is similar to classical fractal dimension, but it focuses on the dominant regions of the distribution while ignoring low-probability areas such as outliers.
>
> > 5. the notation gen is also used before it is defined.
>
> We have revised this in the manuscript. Thank you for pointing it out.
>
> > 6. Can you explain what is the exact role of the neighbourhood U_k in Assumption 3?
>
> The open neighbourhood $U_k$ in Assumption 3 is introduced solely to guarantee that $F_k$, $F_k^*$ and the loss map are differentiable with uniformly bounded gradients on a region containing all embedding points and their small perturbations, so that we can use one Lipschitz constant to uniformly bound how much the loss can change when the embeddings shift, which is what we need for the argument that links Wasserstein distance to the risk difference.

---

> > ### Author Response · Authors · 2025-11-27
> >
> > > 7. In what cases can the intrinsic dimension of the embedding distribution be much smaller than the number of neurons in the layer.
> >
> > In deep networks, it is widely observed that the intrinsic dimension of embeddings is often far smaller than the number of neurons [2]. During training, networks compress data into a low-dimensional “semantic manifold,” with only a few degrees of freedom effectively utilized. If the intrinsic dimension approaches the number of neurons, the data would be effectively unstructured in high dimensions, usually corresponding to high noise or random labels, which is inconsistent with good task performance.
> >
> > > 8. Can you produce the same plot as Fig. 3, but using the number of neurons in the intermediate layer instead of the intrinsic dimension estimator?
> >
> > Thank you for the suggestion. We believe this plot would not be informative, because the number of neurons in the intermediate layer is a fixed architectural hyperparameter and does not reflect the learned geometry of the representation. Our analysis focuses on intrinsic dimension because it captures meaningful variations in the learned representation, whereas architectural width alone does not.
> >
> > > 9. Can you comment on why the reported intrinsic dimensions are quite small on Fig. 3?
> >
> > The small intrinsic dimensions are consistent with many deep representation learning studies. Learned embeddings often concentrate on a low-dimensional semantic manifold, so the effective degrees of freedom in the final layer are much smaller than the ambient dimension. This phenomenon also appears in large models. Extremely high intrinsic dimension would correspond to near-isotropic noise, which is incompatible with good classification performance [2]. Furthermore, Appendix C.5 shows results for large pre-trained models, which confirm that even in these models the embeddings remain relatively low-dimensional.
> >
> > > 10. In addition to Figure 3, could you compute correlation coefficients (eg, Kendall) to get a more precise understanding of the observed correlation?
> >
> > Following the reviewer’s suggestion, we have added correlation coefficients to complement the scatter plots and provide a more precise quantification of the observed relationships.
> >
> > **Reference**
> >
> > [1] Weed, Jonathan, and Francis Bach. "Sharp asymptotic and finite-sample rates of convergence of empirical measures in Wasserstein distance." Bernoulli 25.4A (2019): 2620-2648.
> >
> > [2] Ansuini, Alessio, et al. "Intrinsic dimension of data representations in deep neural networks." Advances in Neural Information Processing Systems 32 (2019).

---

> > > ### Comment · Reviewer_dY4n · 2025-11-27
> > > **Thank you for your answer**
> > >
> > > Thank you very much for your detailed answer.
> > >
> > > After reading it, I have the following questions.
> > >
> > > 1. **About the comparison with Hoeffding inequality:** I understand that you bound does not depend on the uniform bound $B_\ell$ on the loss. Can you then comment on why assumption 5 is needed to your analysis, by looking quickly at the proofs I do not see where this assumption is needed. More generally, I think section 3.5 would need some additional discussion / justification of the assumptions.
> > > Moreover, I agree with you that $B_\ell$ can be very large in many applications, but what if we have a classification problem and we use the 01 loss (or a more regular surrogate, to fit with your assumptions). Can we then still argue that your approach is better because $B_\ell$ is too big?
> > >
> > > 2. **About the predictor depending on the dataset:** From your answer, I understand that you evaluate the embeddings using a validation set rather than the training set, but does it mean that you do not compute the generalization error but rather something like validation error - test error? It is my understanding that, in your theory,  the dataset used to compute the empirical risk has to be the one that we use to compute the embeddings, am I correct?

---

> > > > ### Author Response · Authors · 2025-11-29
> > > >
> > > > Thank you very much for your careful comments. We have re-examined the entire manuscript and carefully reviewed the role of each assumption in the proofs. We confirm that **Assumption 5 is not used anywhere in the analysis**, it has been removed now, and we have reorganized the assumptions and their explanations accordingly.
> > > >
> > > > To avoid confusion, we clarify all situations in the paper where i.i.d. samples are required:
> > > >
> > > > (1) analyzing the properties of the empirical and population embedding distributions, including their dimension and the Wasserstein distance between them;
> > > >
> > > > (2) applying Hoeffding’s inequality to control the empirical noise in **Step 2 of Proposition 2** in Appendix.
> > > >
> > > > Regarding (1), directly using training-set embeddings to analyze the empirical distribution is not rigorous, because the predictor induces coupling with the training data (i.e., the embedding distribution is affected by the training process), and thus the embeddings are no longer i.i.d. This can lead to systematic bias in the estimated  dimension and Wasserstein distance (e.g., underestimating the dimension). To ensure statistical validity, we instead use embeddings from a **validation set** that is independent of training. Importantly, **the definition and computation of the generalization gap remain unchanged**: it still uses training loss and test loss, and the train/test split is not altered.
> > > >
> > > > Regarding (2), the empirical noise refers to the residual between the Bayes predictor and the ground-truth label. Because the Bayes predictor does not depend on any particular training sample, applying the same deterministic function to i.i.d. data yields i.i.d. residuals, making Hoeffding’s inequality applicable for controlling the deviation between their empirical mean and expectation.
> > > >
> > > > Finally, concerning your suggestion about using Hoeffding’s inequality directly to derive a generalization bound when the loss is bounded: this is not rigorous because the trained predictor is data-dependent. As a result, the per-sample training losses are not independent, violating the independence assumption required by Hoeffding. Therefore, Hoeffding cannot be directly used to control the deviation between empirical risk and population risk in this setting.
> > > >
> > > > We sincerely thank you again for your insightful questions and suggestions, which have greatly improved the clarity and quality of our manuscript. If we have made any mistakes, we would greatly appreciate it if you could kindly point them out.

---

### Official Review · Reviewer_mmHw · 2025-11-01

**Soundness:** 2
**Presentation:** 3
**Contribution:** 3
**Rating:** 4
**Confidence:** 4

**Summary:**

This paper presents an exciting argument that the generalization error of deep learning models can be bounded by a combination of the intrinsic dimension of the data at a given layer of the model and its Lipschitz constant at that layer. The authors present a bound from a 2019 paper by Weed & Bach, and point out that the last layer creates a simplification with respect to the Lipshiptz constant. Additionally, the authors present a bound that depends instead on the intrinsic dimension d_k at a given layer of the network. The authors present experiments that attempt to verify that the bounds dependent on the intrinsic dimension are applicable to neural network embeddings, resulting in a series of plots that demonstrate a correlation between intrinsic dimension and a Wasserstein distance (between a empirical and population distribution of embeddings), and Wasserstein distance and generalization gap (difference between test and train error). I was really excited to read this paper, but many of the details of the argument were not clear in the main text leading to my current reservations. I would love to be convinced.

**Strengths:**

- The paper presents a very important line of research for the community: how is the distribution of embeddings related to the generalization performance of a model?
- The approach of breaking a model into an “encoder” and a “tail” network at any given layer is interesting and widely applicable.
- The relationship between Wasserstein distance of an embedding and the generalization gap is interesting.

**Weaknesses:**

- The authors are missing a large body of literature related to predicting generalization performance of models. Authors should address why the 2020 NeurIPS challenge on Predicting Generalization in Deep Learning is not applicable, or unsuitable for this study. Some of the same models are included, but importantly at various levels of training (and test) performance.
- The terminology used in the paper is not consistent. The generalization gap can be approximated as the difference between the train and test error. However, the authors seem to refer to this as the generalization error in the start of the paper. The generalization error can be measured using test accuracy or test loss, i.e. samples that are not in a limited training set. I believe this is an important distinction, because if a model is poorly trained, the gap between train and test may be 0, which is not a very useful. This assumption should at minimum be specified.
- In 5.1, it appears the experiment is only able to demonstrate log-linear correlation between intrinsic dimension and the Wasserstein distance between two independent samples of embeddings, with other terms ignored.
- In 5.2, the Wasserstein distance is instead computed between train and test embedding distributions. Moreover, I believe there is a methodological flaw in the evaluation shown in Figure 3. Fitting curves and R^2 values should be computed for each model independently.
- The main result in the paper seems to based on replacing d in the bound from Weed and Bach with d_k, the intrinsic dimension computed by MLE. I could be wrong but I’m not finding adequate justification for this in the main body of the paper. It is also not clear why delta (1-delta representing a probability) is introduced in Theorem 4.1.

**Questions:**

- Why are two independent draws of embeddings at a given layer suitable for representing the “empirical” and “population” distribution of embeddings?
- How are the distributions of embeddings at a given layer learned or summarized? They are almost surely not Gaussian and cannot be summarized with a multi-variate mean and std alone, so I am a bit surprised.
- Most of the results seem to be focused on the last layer, what about results on the interior layers?
- How is the Bayes predictor used in the empirical experiments, if at all?
- In Line 219, what is meant exactly by “[the Bayes predictor] returns the best possible prediction at each embedding” ?
- Provide a reference for the spectral norm of Linear+ReLU layers bounding the layer’s Lipshitz constant.
- Line 236, McDiarmid’s inequality?

Possible improvements:
- One of the interesting arguments in the paper is that the intrinsic dimension of embeddings at any layer, together with the Lipschitz constant, can be used to bound the generalization gap of the model. Empirical results evaluating this behavior as a function of layer would be interesting and improve the experimental validation portion of the paper.
- The ability to predict the test performance (e.g. given the train performance) of a model, such as in the setup of the the aforementioned PGDL challenge, would increase the impact of this work. I suggest the authors to review the metrics and criteria used in similar work to validate their cited bounds.
- A larger portion of the paper is dedicated to definitions and assumptions, but there is practically no explanation or proof sketch for Theorem 4.1. It would be very useful to explain how Eqn 1 is meaningfully different from the eqn on Line 191 if the leading term indeed dominates the behavior.

---

> ### Author Response · Authors · 2025-11-27
>
> > 1. The authors are missing a large body of literature related to predicting generalization performance of models. Authors should address why the 2020 NeurIPS challenge on Predicting Generalization in Deep Learning is not applicable, or unsuitable for this study.
>
> We thank the reviewer for pointing out the PGDL literature. The PGDL benchmark is indeed important for predicting generalization performance, but its goals and methods differ from ours and cannot directly replace our study.
>
> While both PGDL and our work focus on model generalization, PGDL metrics that use hidden-layer representations (e.g., Consistency of Representations, Robustness of Representations, Separability of Representations) rely on labels or loss information. This prevents their application in scenarios such as pretraining or unsupervised learning, where labels are unavailable [1]. In contrast, our metrics are entirely based on the distributional properties of embeddings and do not require labels, making them broadly applicable. Importantly, our main conclusion is that analyzing the dimension of the last-layer embeddings alone suffices to predict overall model generalization, without any label information, this is a key distinction and advantage.
>
> Regarding the reviewer’s possible concern about why we did not analyze the specific models from PGDL, their architectures are standard deep learning models. We additionally provide results on large-scale pretrained models in Appendix C.5, showing that our conclusions hold for large models and datasets. Importantly, our theoretical results are not tied to any particular model architecture, and the current experiments already cover a diverse range of model scales and dataset sizes.
>
>
> > 2. Terminology of generalization error vs. generalization gap
>
> We thank the reviewer for pointing this out. We have revised the manuscript to consistently distinguish between generalization error (measured on unseen test samples) and generalization gap (difference between training and test errors).
>
>
> > 3. In 5.1, it appears the experiment is only able to demonstrate log-linear correlation between intrinsic dimension and the Wasserstein distance between two independent samples of embeddings, with other terms ignored.
>
> The purpose of analyzing the bound is to understand how embedding dimension affects generalization. Our theoretical results indicate that embedding dimension primarily influences generalization via its effect on the Wasserstein distance between empirical and population distributions. Therefore, our experiments focus on verifying whether embedding dimension, Wasserstein distance, and generalization error are indeed significantly correlated.
>
> We acknowledge that several terms in the bound, such as the network Lipschitz constants, are difficult to estimate for complex networks. Nonetheless, the observed strong correlations among embedding dimension, Wasserstein distance, and generalization error indicate that these uncomputable constants do not dominate the bound. If they did, the changes in generalization error would not correlate significantly with the changes in embedding dimension or Wasserstein distance.
>
>
> > 4. In 5.2, the Wasserstein distance is instead computed between train and test embedding distributions. Moreover, I believe there is a methodological flaw in the evaluation shown in Figure 3. Fitting curves and R^2 values should be computed for each model independently.
>
> We revised the experiments following reviewer dY4n’s suggestion. We now use the validation set as the empirical distribution and the test set as the population distribution, ensuring that validation embeddings are iid.
>
> For Figure 3, aggregating results across different models and classes demonstrates the generality of our findings and the effect of model architecture. We also provide per-class results for the same model in **Appendix C.4**, which remain consistent.

---

> > ### Author Response · Authors · 2025-11-27
> >
> > > 5. The main result in the paper seems to based on replacing d in the bound from Weed and Bach with d_k, the intrinsic dimension computed by MLE. I could be wrong but I’m not finding adequate justification for this in the main body of the paper. It is also not clear why delta (1-delta representing a probability) is introduced in Theorem 4.1.
> >
> > We provide a formal definition of intrinsic dimension in Section 3.4. Specifically, we define $d_k$ in the bound using the Upper Wasserstein Dimension, which measures the effective complexity of the main part of the data distribution. Conceptually, this is similar to classical fractal dimension but focuses on the dominant regions of the distribution while ignoring low-probability areas such as outliers.
> >
> > The parameter δ in Theorem 4.1 arises from concentration inequalities that control deviations between Wasserstein distance and empirical noise (see **Proposition 2 in the Appendix**). Specifically, δ is introduced via McDiarmid’s and Hoeffding’s inequalities to provide a high-probability guarantee: with probability at least 1−δ, the deviation does not exceed a certain bound. This ensures that the bound is statistically reliable and robust, providing a formal confidence level for the deviation between empirical estimates and the true distribution.
> >
> >
> > > 6. Why are two independent draws of embeddings at a given layer suitable for representing the “empirical” and “population” distribution of embeddings?
> >
> > Using independent draws of embeddings to approximate empirical and population distributions is a common practice in the field. In our framework:
> >
> > (1) Empirical distribution: We treat it as the distribution underlying the observed data. Embeddings from a validation set, which is independent of the training process, serve as a reasonable approximation. This ensures that the embeddings are i.i.d. and independent of the predictor, satisfying the conditions for empirical estimation.
> >
> > (2) Population distribution: The test set is treated as unseen data, effectively representing the true data distribution from which samples are drawn.
> >
> > In short, validation embeddings approximate the empirical distribution, and test embeddings approximate the population distribution. Both are practical, widely used approximations that allow us to analyze embedding distributions in a statistically sound manner.
> >
> >
> > > 7. How are the distributions of embeddings at a given layer learned or summarized? They are almost surely not Gaussian and cannot be summarized with a multi-variate mean and std alone, so I am a bit surprised.
> >
> > We do not rely on traditional statistics such as mean or standard deviation, because these cannot capture the complex geometric structure of embedding spaces. Instead, we use:
> >
> > (1) Wasserstein distance: Quantifies the geometric difference between empirical and population distributions. For a fixed sample size, a smaller Wasserstein distance indicates that the embedding distribution is simpler and easier to learn, as the empirical samples closely reflect the population distribution.
> >
> > (2) Intrinsic dimension: Measures the effective degrees of freedom of the embedding distribution. It directly quantifies the complexity of the distribution and captures the richness of the representation space.
> >
> > Together, these measures provide a more accurate and meaningful characterization of embedding distributions than conventional statistics, and better capture how embedding geometry influences generalization performance. Importantly, this analysis does not assume any specific form of the data distribution and can be applied to data from arbitrary distributions.
> >
> > > 8. Most of the results seem to be focused on the last layer, what about results on the interior layers?
> >
> > We focus on the last layer for two reasons:
> >
> > (1) It controls for differences in network width and depth across architectures. Last-layer widths are fixed (determined by the number of classes), whereas earlier layers vary in width and relative depth, introducing additional variability.
> >
> > (2) It minimizes the effect of the network’s Lipschitz constant, making the relationship between embedding dimension and generalization error more pronounced.
> >
> > We also provide results for intermediate layers in **Appendix C.6**. For ResNet152, embedding dimension, Wasserstein distance and generalization error remain highly correlated across layers. Notably, correlations between dimension/Wasserstein distance and generalization strengthen in deeper layers, consistent with theoretical predictions: deeper layers effectively reduce the Lipschitz constants, amplifying the effect of intrinsic dimension.

---

> > > ### Author Response · Authors · 2025-11-27
> > >
> > > > 9. How is the Bayes predictor used in the empirical experiments, if at all?
> > >
> > > The Bayes predictor represents a hypothetical ideal model that can perfectly predict outputs for any input. In classification, it converts discrete labels into continuous outputs and is assumed to be Lipschitz-continuous. This allows decomposition of the loss’s Lipschitz constant into contributions from the model and the Bayes predictor (**Lemma 2 in Appendix**).
> > >
> > > In practice, the Bayes predictor is theoretical and is not analyzed in empirical experiments.
> > >
> > > > 10. In Line 219, what is meant exactly by “[the Bayes predictor] returns the best possible prediction at each embedding” ?
> > >
> > > The Bayes predictor provides the optimal output for each input. Conceptually, it allows analysis of generalization error as the difference between the trained model and an ideal model. Continuous outputs facilitate derivative-based analysis, although small unavoidable deviations exist (e.g., true labels [0, 1, 0, 0] versus Bayes outputs [0.12, 0.95, 0.21, 0.13]).
> > >
> > >
> > > > 11. Provide a reference for the spectral norm of Linear+ReLU layers bounding the layer’s Lipshitz constant.
> > >
> > > We have added the reference in the main text [2].
> > >
> > > > 12. Line 236, McDiarmid’s inequality?
> > >
> > > In Appendix Proposition 2, we use McDiarmid’s inequality to bound deviations between Wasserstein distance and its expectation. This requires assuming bounded differences: changing any single embedding changes the Wasserstein distance by a bounded amount, allowing the inequality to be applied.
> > >
> > >
> > > > 13. A larger portion of the paper is dedicated to definitions and assumptions, but there is practically no explanation or proof sketch for Theorem 4.1.
> > >
> > > We have added detailed explanations in Theorem 4.1, clarifying the contributions of individual terms.
> > >
> > >
> > > **Reference**
> > >
> > > [1] Jiang, Yiding, et al. "Methods and analysis of the first competition in predicting generalization of deep learning." NeurIPS 2020 Competition and Demonstration Track. PMLR, 2021.
> > >
> > > [2] Bartlett, Peter L., Dylan J. Foster, and Matus J. Telgarsky. "Spectrally-normalized margin bounds for neural networks." Advances in neural information processing systems 30 (2017).

---

### Official Review · Reviewer_NeUQ · 2025-11-07

**Soundness:** 2
**Presentation:** 2
**Contribution:** 2
**Rating:** 2
**Confidence:** 3

**Summary:**

This paper proposes a novel, representation-centric generalization error bound for neural nets. Unlike traditional parameter-based bounds, the authors link generalization to the geometric properties of the learned embeddings.

The main theoretical claim is that the generalization error at any given layer k is controlled by two key factors:
1. the intrinsic Dimension of the embedding distribution at layer k.
2. Lipschitz Constants of the downstream network (from layer k to the final output). These measure how much perturbations in the embedding space are amplified in the loss.

A key insight is that this bound simplifies at the final layer. Here, the downstream map is the identity (i.e. Lipschitz constant 1) and the generalization error hence is driven by the intrinsic dimension of the final-layer embeddings and a data-dependent notion of smoothness (via a certain Bayes predictor) .

The authors also provide empirical support for their theory.

**Strengths:**

- This paper provides an interesting theoretical bridge between representation geometry (via intrinsic dimension) and generalization.
- The central idea of combining Wasserstein convergence rates (Weed & Bach, 2019) together with a perturbation analysis based on the network-dependent Lipschitz constants appears novel.
- Overall, the paper is well-written, with one exception (see Weaknesses).

**Weaknesses:**

- It is not clear what is formally meant by “intrinsic dimension”, which is defined via “effective dimension” of $P_k^Z$, but never formally discussed. Initially, I assumed that you mean the cardinality of the support, put it appears that your notion of dimension can have fractional values. Since this quantity is a critical component of the analysis, it should be crystal clear to the reader what $d_k$ actually is. Likewise, it should be clear what are the limitations of practically estimating the $d_k$’s.
- I do not quite understand why the correlation between the bound and the generalization gap is not shown alongside the empirical results. The presented results show that certain components of the bound correlate with the generalization gap, but this is ultimately not sufficient to claim that the bound provides an compelling explanatory theory of generalization.
- Practical Computability of the Bound: as with most work in this area, the derived bound is not practically computable. It contains several terms ($L_k(F)$, $L_k(F^*)$, $D_k$, etc.) that are either data-dependent and unobservable (the Bayes predictor's Lipschitz constant) or computationally hard to estimate for complex, modern architectures. The value of the bound is therefore mostly conceptual.
- Adding to this, For intermediate layers, $\bar L_k$ is estimated via spectral-norm products proxy; this is an upper bound and can severely overestimate sensitivity.

**Questions:**

- What is the definition of the intrinsic dimension? Are there well-known practical difficulties to estimating it?
- Does the actual bound itself meaningfully correlate with the generalization gap?
- It seems that across your experiments, you sometimes obtain intrinsic dimensions $>2$ (even 5-6 in case of MNIST), which corresponds to your generalization bounds decaying more slowly than $\sim 1/\sqrt{n}$. How can this be reconciled with classical VC theory which states that learning is at most as slow as $1/\sqrt{n}$ (as soon as the number of samples exceeds the VC dimension)?
- For CIFAR-10/CIFAR-100, when you average the intrinsic dimension across classes in your experiments, do you obtain something greater or smaller than $2$?
- Regarding sec 5.4.: did you also compute the correlation of the whole bound with the generalization error when varying the width? It seems that your figure 4 cannot reveal whether the generalization bound admits the correct tendency under changing the width.

---

> ### Author Response · Authors · 2025-11-27
>
> >1. Practical Computability of the Bound
>
> We acknowledge that some terms, like the Lipschitz constant, are hard to estimate for complex networks. For ReLU networks, using the product of spectral norms provides only an upper bound. Thus, our bound cannot yield a concrete numeric value.
> However, our goal is not to provide a tighter bound than previous work, but to explain why smaller embedding dimensions often correlate with better generalization [1]. The contributions of this work are:
>
> (1) Introducing a generalization bound from the perspective of embeddings.
>
> (2) Explaining that embedding dimension mainly affects the Wasserstein distance between the empirical and true distributions, which in turn influences generalization error.
>
> Although the bound is not directly computable, our experiments show that actual generalization error correlates significantly with estimable quantities like embedding dimension and Wasserstein distance, suggesting that other terms in the bound are not dominant.
>
> We also show in **Appendix C.5** that our results hold for larger models and datasets.
>
> > 2. Estimation of intermediate layer Lipschitz constants
>
> Yes, this is correct. Currently, there is no better practical estimation method.
>
> > 3. Definition and estimation of intrinsic dimension
>
> We provide a formal definition in Section 3.4. There, we define intrinsic dimension using the Upper Wasserstein Dimension, which quantifies the complexity of the main part of the data distribution. Conceptually, this approach is similar to classical fractal dimension, but it focuses on the dominant regions of the distribution while ignoring low-probability areas such as outliers.
>
> Estimating intrinsic dimension can be challenging with small sample sizes [2]. However, our experiments show that even with only 500 samples, the estimated dimension exhibits significant correlation with generalization error, a sample size easily met in modern deep learning. High dimension can also make estimation difficult, but neural network embeddings are typically low-dimensional [1], so this issue is limited.
>
> To ensure robustness, we used three methods (MLE, MOM, TLE; see **Appendix C.2**), all producing consistent results.
>
> > 4. Relation to VC theory
>
> VC dimension measures the complexity of a function class, i.e., the model’s expressivity. In contrast, our intrinsic dimension is a property of the embeddings of a single model on a dataset. These are two completely different concepts.
>
> Consequently, even very large models may have extremely high VC dimension, making VC-based generalization bounds vacuous. In contrast, intrinsic dimension remains moderate, as it reflects the complexity of the data distribution rather than model size. Therefore, our bound remains meaningful and non-vacuous even for large models.
>
> > 5. For CIFAR-10/CIFAR-100, when you average the intrinsic dimension across classes in your experiments, do you obtain something greater or smaller than 2?
>
> We performed additional experiments (**Appendix C.3**). For CIFAR-10, per-class dimensions are similar, so averaging preserves trends with generalization. For CIFAR-100, class dimensions vary widely; averaging across classes gives noisy results. Combining all classes into a single computation better reflects overall data complexity, yielding significant correlations for both CIFAR-10 and CIFAR-100.
>
> > 6. Effect of network width on the bound
>
> Due to the presence of uncomputable constants in the bound, it is difficult to directly analyze the correlation between the bound and actual generalization error. The main purpose of Section 5.4 is to explain why reducing network width decreases intrinsic dimension, but does not necessarily improve generalization performance. Narrower networks increase the network’s Lipschitz constant, which counteracts the potential benefit from lower intrinsic dimension.
>
>
> **Reference**
>
> [1] Ansuini, Alessio, et al. "Intrinsic dimension of data representations in deep neural networks." Advances in Neural Information Processing Systems 32 (2019).
>
> [2] Levina, Elizaveta, and Peter Bickel. "Maximum likelihood estimation of intrinsic dimension." Advances in neural information processing systems 17 (2004).

---

### Author Response · Authors · 2025-11-27
**Global Response**

We thank all reviewers for their constructive feedback. In response, we have made substantial theoretical and empirical improvements throughout the manuscript. The major revisions are summarized as follows:

**1.Large-model and large-dataset validation.**
We added experiments on large pretrained models and ImageNet datasets (**Appendix C.5**). These results confirm that intrinsic dimension and Wasserstein distance remain strongly correlated with generalization even in high-capacity and high-complexity settings.

**2.Layer-wise analysis of embeddings.**
We expanded the empirical evaluation to all intermediate layers (**Appendix C.6**), showing that intrinsic dimension, Wasserstein distance, and generalization error maintain strong correlations across depth, with effects becoming more pronounced in deeper layers.

**3.Dimension dynamics across training/validation/test sets.**
We analyzed how intrinsic dimension evolves across training, validation, and test embeddings (**Appendix C.7**). The dimensions of the three splits remain close to each other, and the  dimension exhibits the non-monotonic pattern, initially decreasing and later increasing.

**4.Clearer definition of intrinsic/effective dimension and refined theory explanation.**
Section 3.4 now includes a more precise definition of intrinsic/effective dimension based on the Upper Wasserstein Dimension, along with detailed explanations of each term in Theorem 4.1 and its role in the bound.

**5.Independence and empirical distribution correctness**
To ensure theoretical validity, all statistical quantities (intrinsic dimension, Wasserstein distance) are now computed from validation embeddings. This guarantees independence between the predictor and the samples used for estimating empirical distributions.

---

### Author Response · Authors · 2025-12-03
**Summary**

We thank the reviewers for their constructive feedback. Their comments have substantially improved the clarity, rigor and empirical breadth of our work. Below we summarize our main contributions and the enhancements made during the rebuttal.

**Core Contribution**

Our paper proposes a representation-centric generalization bound that explicitly links generalization performance to two measurable properties of neural network embeddings:

**1. Intrinsic Dimension of Embeddings**: Determines how fast the empirical embedding distribution converges to its population counterpart in Wasserstein distance. Lower intrinsic dimension yields faster convergence and thus smaller generalization error.

**2. Lipschitz Continuity of the Downstream Network**: Measures how perturbations in embeddings propagate through the tail of the network to affect the loss.

These two factors jointly control the generalization error at intermediate layers. Critically, **at the final embedding layer, the architectural Lipschitz constant vanishes**, making generalization primarily governed by embedding dimension. This gives a theoretical explanation for a widely observed phenomenon in deep learning: **final-layer intrinsic dimension is a strong predictor of generalization performance across models and datasets**.

**Impact and Insight**

Our theory explains why lower-dimensional embeddings empirically correlate with better generalization, and provides actionable insight for practice: **to improve generalization, one should optimize representations toward lower-dimensional final-layer embeddings.**

**Enhancements Made During Rebuttal**

Reviewers asked for broader empirical support, clearer theoretical assumptions, and more nuanced analysis. In response, we made the following substantial improvements:

**1. Large-Scale Experiments Added**: We added results on large pretrained models and ImageNet (Appendix C.5), demonstrating that the dimension–Wasserstein–generalization relationships persist at large scale. This strongly supports the generality of our theory.

**2. Refined Theoretical Assumptions and Exposition**: We clarified the definition of intrinsic dimension, tightened the role of the Bayes predictor, and improved the presentation of the loss sensitivity terms and Wasserstein convergence assumptions to enhance rigor and readability.

**3. Layer-wise and Training-Dynamic Analyses Added**: We expanded analysis of intrinsic dimension, Wasserstein distance, and generalization across layers, architectures, and during training, showing:

* intrinsic dimension does not monotonically decrease during training,

* but its final value remains predictive of generalization, matching our theory.


**Summary**

After the rebuttal, the paper provides:

* a rigorous, explicit generalization bound driven by intrinsic dimension and Lipschitz sensitivity,

* a clean final-layer simplification where generalization is governed mainly by embedding dimension,

* extensive empirical support across datasets, architectures, model sizes, and interventions,

* improved structure and clarity in both the theoretical and empirical sections.

We thank the reviewers again. Their feedback helped significantly strengthen both the theoretical exposition and the empirical validation.

---

### Note · Authors · 2026-01-17

I have read and agree with the venue's withdrawal policy on behalf of myself and my co-authors.